# Mitochondrial Impairment: A Common Motif in Neuropsychiatric Presentation? The Link to the Tryptophan–Kynurenine Metabolic System

**DOI:** 10.3390/cells11162607

**Published:** 2022-08-21

**Authors:** Masaru Tanaka, Ágnes Szabó, Eleonóra Spekker, Helga Polyák, Fanni Tóth, László Vécsei

**Affiliations:** 1ELKH-SZTE Neuroscience Research Group, Danube Neuroscience Research Laboratory, Eötvös Loránd Research Network, University of Szeged (ELKH-SZTE), Tisza Lajos krt. 113, H-6725 Szeged, Hungary; 2Department of Neurology, Albert Szent-Györgyi Medical School, University of Szeged, Semmelweis u. 6, H-6725 Szeged, Hungary; 3Doctoral School of Clinical Medicine, University of Szeged, Korányi fasor 6, H-6720 Szeged, Hungary

**Keywords:** mitochondria, stress resilience, plasticity, stress, kynurenine, Alzheimer’s disease, neurodegenerative, depression, anxiety, psychiatric

## Abstract

Nearly half a century has passed since the discovery of cytoplasmic inheritance of human chloramphenicol resistance. The inheritance was then revealed to take place maternally by mitochondrial DNA (mtDNA). Later, a number of mutations in mtDNA were identified as a cause of severe inheritable metabolic diseases with neurological manifestation, and the impairment of mitochondrial functions has been probed in the pathogenesis of a wide range of illnesses including neurodegenerative diseases. Recently, a growing number of preclinical studies have revealed that animal behaviors are influenced by the impairment of mitochondrial functions and possibly by the loss of mitochondrial stress resilience. Indeed, as high as 54% of patients with one of the most common primary mitochondrial diseases, mitochondrial encephalomyopathy with lactic acidosis and stroke-like episodes (MELAS) syndrome, present psychiatric symptoms including cognitive impairment, mood disorder, anxiety, and psychosis. Mitochondria are multifunctional organelles which produce cellular energy and play a major role in other cellular functions including homeostasis, cellular signaling, and gene expression, among others. Mitochondrial functions are observed to be compromised and to become less resilient under continuous stress. Meanwhile, stress and inflammation have been linked to the activation of the tryptophan (Trp)–kynurenine (KYN) metabolic system, which observably contributes to the development of pathological conditions including neurological and psychiatric disorders. This review discusses the functions of mitochondria and the Trp-KYN system, the interaction of the Trp-KYN system with mitochondria, and the current understanding of the involvement of mitochondria and the Trp-KYN system in preclinical and clinical studies of major neurological and psychiatric diseases.

## 1. Introduction

Mitochondria are double membrane-bound cell organelles abundant in the cytosol of eukaryotes. The most prominent role of mitochondria is the production of high-energy storage molecule adenosine triphosphate (ATP) [1,2,3]. For ultimate energy production, mitochondria employ a variety of metabolic activities, including the tricarboxylic (TCA) cycle, oxidative phosphorylation (OXPHOS), ketogenesis/ketolysis, fatty acid oxidation, and glutamate metabolism [4,5,6,7,8,9]. Each component forms a complex metabolic network and dynamically adapts to the cellular environment to ensure the optimal energy supply. Mitochondrial malfunction can occur due to the defects of proteins directly or indirectly responsible for the OXPHOS or to the dysfunction of cellular mechanisms outside of mitochondria [10,11,12,13]. However, the role of mitochondria is not limited to cellular energy production. Other functions of mitochondria include calcium storage, subcellular signaling such as gene expression, autophagy, and apoptosis, among others [14,15].

Meanwhile, the tryptophan (Trp)–kynurenine (KYN) metabolic system plays a major role in Trp metabolism, as over 95% of Trp catabolizes into nicotinamide adenine dinucleotide (NADH). Accumulating evidence is revealing that the enzymes and the metabolic products of the Trp-KYN system actively influence the metabolism of mitochondria and participates in normal aging in organisms as well as the pathogenesis of mitochondrial diseases, neurodegenerative diseases, and psychiatric disorders [16,17]. The enzymes of the Trp-KYN system are activated by inflammation, oxidative stress, antioxidant system, and downstream bioactive metabolites [18,19].

Normal functions of mitochondria are typically compensated in mitochondrial diseases, a heterogenous group of chronic, genetic, and often inherited metabolic disorders caused by mitochondrial dysfunction, resulting in the impairment of cellular energy production and other crucial mitochondrial functions [20]. The prevalence of inherited mitochondrial diseases is estimated to occur one in 5000 live births, and they are the most common inborn errors of metabolism [21]. Primary mitochondrial disease (PMD) is caused by the pathogenic mutation of mitochondrial DNA (mtDNA) or nuclear DNA (nDNA) encoding either the proteins of OXPHOS or the proteins affecting the energy production of OXPHOS. Secondary mitochondrial diseases (SMDs) can be hereditary, caused by the genes for mtDNA transcription or expression, homeostasis, or metabolism [22]. Furthermore, mitochondrial dysfunction can be caused by acquired multifactorial diseases such as diabetes, cancer, heart or kidney disease, or neurodegenerative diseases [23,24].

Mitochondrial malfunction occurs in other conditions including normal aging in organisms, neurodegenerative diseases, and psychiatric disorders. Age-related physiological changes are strongly associated with mitochondrial malfunction with decreased mtDNA volume and mitochondrial integrity, which results from cumulative damage to mtDNAs by reactive chemical species [25]. Mitochondrial dysfunction also occurs in most neurodegenerative diseases such as Alzheimer’s disease (AD), Parkinson’s diseases (PD), Huntington’s disease (HD), Friedreich’s ataxia (FA), and amyotrophic lateral sclerosis (ALS) [26]. Psychiatric disorders include mood disorders such as major depressive disorder (MDD) and bipolar disorder (BD), schizophrenia (SCZ), autism spectrum disorder (ASD), and attention-deficit hyperactive disorder (ADHD) [27].

This review discusses the functions of mitochondria, the Trp-KYN system, the interaction of the Trp-KYN system with mitochondria, the mitochondrial environment upon the activation of the Trp-KYN system, and the link to neuropsychiatric presentation in clinical and preclinical settings in search for possible diagnostic biomarkers and novel interventional targets for mitochondria-associated diseases.

## 2. Mitochondria in the Central Nervous System

The brain accounts for only 2% of the body weight; however, it consumes as much as 20% of body’s total oxygen supply. An estimated number of one to two million mitochondria is present per single neuron in the human substantia nigra [28]. Mitochondria take responsibility for the production of cellular energy and the proper conduction of neural circuits in the nervous system [29,30,31,32,33]. Mitochondria are multifunctional organelles maintaining calcium homeostasis and signaling to other organelles in the cell as well as with other mitochondria at distance [34,35]. Furthermore, mitochondria are highly plastic in morphology, functions, and cell cycle, depending on the tissue type and the need of cells [36]. Mitochondria can be even transferred from cell to cell [37].

### 2.1. Mitochondrial Bioenergetics

Glucose, other sugars, and some amino acids are broken down in the cytosol to three-carbon molecule pyruvate which transfers into the mitochondria. Pyruvate is degraded to two carbon molecule acetyl coenzyme A (acetyl-CoA) which enters the second stage of cellular respiration, the TCA cycle that takes place in the matrix of mitochondria. Initially, Szent-Györgyi reported a cyclic chemical reaction between a four-carbon molecule (oxaloacetate) and two four-carbon molecules (fumarate and L-(-)-malate (the Szent-Györgyi cycle)). Later, Krebs revealed larger cyclic biochemical reactions in which a two-carbon molecule “triose” bonds with oxaloacetate to form a six-carbon molecule citrate, which is then oxidized to a five-carbon molecule (alpha-ketoglutarate) and four-carbon molecules (succinyl-CoA, succinate, fumarate, and malate), thus forming the TCA cycle (the Krebs cycle) [38]. “Triose” was eventually identified as a product of pyruvate and coenzyme A, acetyl-CoA [39,40]. The TCA cycle employs eight different enzymes, reproducing one molecule of oxaloacetate, two molecules of carbon dioxide, water, three molecules of NADH, and one molecule of flavin adenine dinucleotide (FADH_2_) and guanosine triphosphate (GTP). GTP is readily converted to ATP. In the TCA cycle, most of the high-energy storage molecule ATP is consumed by NAD^+^ and FAD to form NADH and FADH_2_ [29] (Figure 1a). The NAD^+^ excess has been reported to improve mitochondrial function and thus prolong the life span of mice [30].

Of note, 16–18 carbon chain fatty acids transported by plasma albumin diffuse into the cytosol using a protein transporter. Consuming ATP, fatty acid is transformed to acyl coenzyme A (acyl-CoA) that crosses the inner membrane of mitochondria by carnitine-acyl-CoA transferase. The beta-oxidation takes place in the mitochondrial matrix in which acetyl-CoA, water, and five ATP molecules are produced by shortening two carbon chains until an acyl-CoA molecule is reduced to an acetyl-CoA molecule [31].

Amino acids are recycled to produce new proteins, but when they are in excess, or cells are under starvation, amino acids are catabolized to supply energy. All essential amino acids except histidine, alanine, and cysteine (Cys) are involved in mitochondrial metabolic pathways. All essential amino acids are converted to pyruvate in the cytosol, which enters to mitochondria to fuel in the TCA cycle [36].

NADH and FADH_2_ transfer their energy to the third stage of cellular respiration OXPHOS consisting of the electron transport, chemiosmosis, and ATP synthesis. The electron transport chain (ETC) generates a proton (H^+^) gradient across the inner membrane and the subsequent return of the H^+^ to the matrix produces ATP from ADP by ATP synthetase. The ETC is a group of protein complexes composed of the NADH coenzyme Q reductase (Complex I), coenzyme Q, succinate dehydrogenase (Complex II), cytochrome bc1 complex (Complex III), and cytochrome c oxidase (Complex IV). NADH donates an electron to Complex I, generating three protons, while FADH_2_ donates an electron to Complex II, generating two protons. ATP synthetase (Complex V) utilizes a proton gradient across the inner membrane to synthesize ATP from ADP and inorganic phosphate (Pi) [32]. In general, three to four protons are required to produce one ATP with 42% efficiency of energy conservation [33]. However, the cellular energy production can be altered under stressful condition or pathological processes according to the availability of substrates, enzyme activity, mitochondrial and cellular conditions, and adjacent biosystems including Trp-KYN metabolic system.

### 2.2. Other Mitochondrial Functions

Mitochondria play an important role in cellular calcium homeostasis. The concentration of calcium ions in the intermembrane space is the same as that in the cytosol due to the high permeability of the outer mitochondria membrane. A higher concentration of mitochondrial calcium ions enhances ATP production; however, severe calcium overloads are associated with pathological conditions [34,35,41].

Mitochondria constantly communicate with other cellular organelles such the nucleus, the ER, lysozymes, and peroxisomes. The coordinated interaction of mitochondrial and nuclear factors is required for mitochondrial gene expression offered by mitochondrial ribonuclease P, ribosomal RNAs, transfer RNAs, introns, and a protein [42]. The nucleus sends signals to the mitochondria via anterograde regulation to modulate mitochondrial biogenesis upon stressful events. On the other hand, mitochondria constantly transmit information on mitochondrial status and cellular stress to the nucleus by retrograde signaling [43].

Mitochondria and the ER are at a close contact through the mitochondria-associated membrane to exchange information on energy production, calcium homeostasis, lipid transport, and apoptosis [44]. Lysosomes interact with mitochondria to transport amino acids, lipids, and calcium ions [45]. Mitochondria and peroxisomes function in concert in fatty acid metabolism. Mitochondria degrade long-chain fatty acids to supply acetyl-CoA and produce ATP, while peroxisome performs beta-oxidation to generate hydrogen peroxide and anabolic metabolic metabolism such as plasmalogen and bile acid synthesis [46].

Mitochondria undergo division during mitosis, dividing equally between the cell soma to daughter cells in interaction with the ER and cytoskeleton [47]. The morphology, functions, and dynamics of mitochondria change upon tissue differentiation [48]. Mitochondria constantly divide and fuse, controlling their morphology and functions. The fusion takes place by initially merging the outer membrane and subsequently the inner membrane of two mitochondria. The continuous events of fusion and division generate mitochondrial networks [49]. Mitophagy refers to mitochondrial autophagy in which double-membraned vesicle autophagosomes deliver mitochondria to lysosomes for destruction. Mitophagy is induced by prolonged fission, promoting the repair process, but may lead to mitochondrial degradation. MicroRNAs play an important role in regulation of protein expression responsible for autophagy [50].

Mitochondria also induce an immune response via the activation of the mitochondrial antiviral signaling protein which leads to the secretion of cytokines via the virally infected cells [51]. Furthermore, mitochondria induce mitochondrial apoptosis through mitochondrial outer membrane permeabilization which leads to the disruption of mitochondrial outer membrane and the release of intermembrane space proteins such as cytochrome c [52]. Therefore, mitochondria impairment may lead to multifarious consequences from ion homeostasis to entire organismal levels. 

## 3. The Tryptophan–Kynurenine Metabolic System

The aromatic amino acid L-Trp is an essential component for the biosynthesis of proteins and a substrate for the production of neurotransmitters and hormones [53]. Acute Trp deprivation increases pain sensitivity, motor activity, acoustic startle, and aggression, while chronic Trp deficiency induces ataxia, cognitive impairment, and dysphoria [54,55]. A meta-analysis showed decreased levels of Trp in blood samples of patients with MDD and BD [56]. Trp is catabolized in the serotonin (5-HT) and KYN metabolic system. More than 95% of L-Trp enters the KYN system, producing several biomolecules. The main enzymes of the KYN system are tryptophan 2,3-dioxygenase (TDO), indoleamine 2,3-dioxygenases (IDOs), kynurenine formamidase (KFA), kynurenine 3-monooxygenase (KMO), kynurenine aminotransferases (KATs), and kynureninase (KYNU). The main metabolites are L-KYN, kynurenic acid (KYNA), 3-hydroxy-L-kynurenine (3-HK), quinolinic acid (QUIN), and NAD^+^. Those metabolites possess a wide range of biological properties such as oxidative, antioxidative, neurotoxic, neuroprotective, cognition-enhancing and impairing, and/or immunomodulating properties, and have attracted growing attention as potential biomarkers, therapeutic targets, cross-species markers, and indicators for environmental resilience [19,57,58] (Figure 2).

### 3.1. Tryptophan 2,3-Dioxygenase, Indoleamine 2,3-Dioxygenases, and Kynurenine Formamidase

TDO and IDOs are heme-containing enzymes that catalyze the oxidation of L-Trp to N-formyl-L-kynurenine. This is the first rate-limiting step of the Trp-KYN system, which regulates the systemic level of Trp in the body. IDO isoform 1 (IDO1) also catalyzes a stereoisomer D-Trp which is a product of the gut microbiome. TDO is a cytosolic enzyme encoded by the gene *tdo2*, stimulated by the stress hormone cortisol and the downward metabolite 3-HK [17,59]. Human *tdo2* gene polymorphisms are associated with ADHD, Tourette syndrome, MDD, ASD, and SCZ [60,61]. *tdo2*^−/−^ mice exhibited less anxious behavior, increased exploratory activities, and increased cognitive function, with increased concentration of Trp, 5-HT, 5-hydroxyindoleacetic acid, and/or KYN (in the plasma, hippocampus, or midbrain) [62,63]. However, it was later reported that anxiolytic and exploratory behaviors are less prominent in *tdo2*^−/−^ mice [64,65]. Thus, the behaviors of *tdo2*^−/−^ mice remain inconclusive.

IDOs are cytosolic enzymes with two isoforms. IDO1 is encoded by the gene *ido1* and expressed in various parts of the body, including the brain, while IDO isoform 2 (IDO2) is encoded by the gene *ido2* and expressed widely in tissues such as kidney, liver, or antigen-presenting cells [66,67]. The two isoforms differentiate in kinetics, substrate specificity, and function. IDOs are upregulated by pro-inflammatory cytokines and lipopolysaccharide but are downregulated by anti-inflammatory cytokines and the antioxidant enzyme superoxide dismutase [17].

*ido1*^−/−^ knockout (KO) mice showed diurnally hypolocomotive behavior with higher brain 5-HT levels and attenuated nociceptive sensation and depressive-like behavior [63,68]. Furthermore, Bacille Calmette-Guérin (BCG) elicited proinflammatory cytokines, but BCG-induced depressive-like behavior was not induced in *ido1*^−/−^ KO mice [69]. However, another study reported that *ido1*^−/−^ KO mice did not exhibit any significantly different behavior, compared to the wild type, in comprehensive behavioral assessments including the domain of cognitive function, negative valence system, motor function, social interaction, and pain sensitivity [70]. *Ido2*^−/−^ KO mice increased exploratory activity during light phase [63]. *ido1* polymorphism is associated with the susceptibility of interferon α treatment-induced depression in patients with chronic hepatitis C [71]. The common polymorphisms of *ido1* and *ido2* genes are associated with the outcome of a selective serotonin reuptake inhibitor (SSRI) citalopram treatment [72].

N-formyl-L-kynurenine hydrolyses to L-KYN spontaneously or enzymatically via KFA. The enzyme is encoded by the gene *afmid*, predominantly cytosolic, expressed in the liver and kidney, and participates in glyoxylate and dicarboxylate metabolism [73]. KFA is stimulated by the proinflammatory cytokine interferon-γ. L-KYN is an antioxidant and an aryl hydrocarbon receptor (AHR) agonist [74]. A meta-analysis showed decreased levels of KYN and an increased KYN/Trp ratio in blood samples of patients with MDD [56]. No study regarding KFA polymorphism or gene KO has been reported (Table 1).

### 3.2. Kynurenine 3-Monooxygenase

KMO is encoded by the gene *kmo* and catalyzes the rate-limiting step of the redox reaction from L-KYN to 3-HK. KMO is located in the outer membrane of mitochondria and is expressed in many tissues of the body, including the brain glial cells and microglia [75]. KMO is stimulated by oxygen molecules, pro-inflammatory cytokines, and the downstream metabolite NADH, while it is inhibited by the superoxide dismutase and anti-inflammatory cytokines [17]. 3-HK generates free radicals to elicit excitotoxic injury. The oxidant molecule 3-HK may function as an antioxidant in certain conditions [76]. A meta-analysis showed a decreased ratio of KYNA/3-HK in blood samples of patients with MDD [56].

*kmo*^−/−^ KO mice have been generated to study the transgenic effects on the metabolites of the Trp-KYN metabolites. The levels of 3-HK were lower in the liver, brain, and plasma; the levels of QUIN were greatly lower in the liver and plasma, while slightly lower in the brain; the levels of KYN, KYNA, and anthranilic acid (AA) were substantially higher, but depending on a tissue; and the levels of NAD^+^ were not different, compared to the wild type [77]. *kmo*^−/−^ KO mice showed lower contextual memory function, more anxious-like behavior, and higher horizontal activity upon a D-amphetamine challenge. The behaviors were associated with the elevated levels of KYNA in the brain, especially in the cerebellum. [78].

A small sample study reported that KMO single-nucleotide polymorphism (SNP) rs1053230 polymorphism was potentially associated with lower CSF KYNA concentrations in SCZ patients [79]. KMO polymorphism has been associated with cognitive dysfunction. The *KMO* rs2275163C>T C (risk) allele was related to the lower cognitive performance in healthy controls, and it is more prominent in SCZ patients. Furthermore, other KMO polymorphism showed a trend effect in cognitive function [80] (Table 1).

### 3.3. Kynurenine Aminotransferases

KATs belong to transferases, specifically transaminases, employing pyridoxal 5’-phosphate (PLP). KATs typically catalyze substrates L-KYN and 2-oxoglutarate to 4-(2-aminophenyl)-2,4-dioxobutanoate and L-glutamate, and then the unstable former product forms KYNA via intramolecular cyclization [81]. Kynurenine–oxoglutarate transaminase 1 (aka KAT I) encoded by the gene *kyat1*, kynurenine/α-aminoadipate aminotransferase (KAT/AadAT, aka KAT II) encoded by the gene *aadat*, kynurenine–oxoglutarate transaminase 3 (KAT III) encoded by the gene *kyat3*, and aspartate aminotransferase (mAspAT, aka KAT IV) encoded by the gene *got2* are isofoms representing KATs. KAT I is located in the cytosol; KAT II is in the inner membrane of mitochondria; KATIII is in the inner membrane of mitochondria and the cytosol; and KAT IV is in the matrix of mitochondria and the plasma membrane [82,83,84,85]. KYNAs are allosterically regulated by α-ketoglutarate in cooperation with KYN [86].

KAT I also catalyzes the reaction of S-substituted L-Cys and H_2_O to a thiol, NH_4_, and pyruvate, as well as the reaction of 3-phenylpyruvate and L-glutamine to 2-oxoglutarate and L-phenylalanine (Figure 1b). 2-oxoglutarate is the same as α-ketoglutarate [82]. Mitochondrial KAT II, KAT III, and KAT IV may compete for the substrate α-ketoglutarate of the TCA cycle, supplying L-glutamate in glutamine metabolism (Figure 1c) [83,84,85]. KAT II also catalyzes the reaction of 2-oxoglutarate and L-2-aminoadipate to 2-oxoadipate and L-glutamate. 2-oxoadipate is an intermediate molecule of L-lysin catabolism, which is further degraded to glutaryl coenzyme A, and finally to acetyl-CoA, forming a side loop of the TCA cycle (Figure 1d) [83]. KAT III catalyzes the reactions of 3-HK and glyoxylate to glycine, H_2_O, and xanthurenic acid (XA); of glyoxylate and L-KYN to glycine, H_2_O, and KYNA; and of an *S*-substituted L-Cys and H_2_O to a thiol, NH_4_^+^, and pyruvate (Figure 1e) [84]. Furthermore, mitochondrial KAT IV catalyzes the reaction of 2-oxoglutarate and L-aspartate to L-glutamate and oxaloacetate, thus bypassing the TCA cycle and participating in essential amino acid aspartate metabolism (Figure 1f) [85]. Thus, KAT enzymes relay molecules between the TCA cycle and glutamate, lysine, and aspartate metabolisms.

KYNA is a receptor antagonist of glutamate receptors including ionotropic α-amino-3-hydroxy-5-methyl-4-isoxazolepropionic acid (AMPA), kainate, and N-methyl-D-aspartic acid (NMDA) receptors [87]. KYNA’s action depends on its concentration at AMPA receptor and cognitive function [88,89]. KYNA’s action at the α-7 nicotinic acetylcholine receptor in vivo remains controversial [90]. KYNA is a G-protein-coupled receptor 35 (GPR35) ligand and an AHR agonist. A meta-analysis showed decreased ratios of KYNA/KYN, KYNA/QUIN, and KYNA/3-HK in patients with MDD; a decreased level of KYNA; and a decreased ratio of KYNA/QUIN in patients with BD [56]. XA is an AHR agonist and a glutamate vesicular transporter (VGLUT) inhibitor and may be a Group II metabotropic glutamate receptor agonist [91,92]. A meta-analysis revealed lower XA levels in the blood of patients with BP and a significantly lower level of XA was observed in the serum of SCZ patients [93,94].

The intracerebroventricular (i.c.v.) administration of KYNA was reported to enhance memory function at low doses and impair it at higher doses in the passive avoidance test of mice [95]. Furthermore, i.c.v. KYNA and synthetic KYNA analogues showed antidepressant-like effects in modified forced swim test of mice [96,97].

*aadat*^−/−^ (aka *kat2*^−/−^) KO mice exhibited transitory hyperlocomotive activity and abnormal motor coordination during postnatal day 17 to 26 with transiently reduced total brain KAT activity and KYNA levels during the first month, which returned to normal levels later [94]. Three-week-old *kat2*^−/−^ KO mice showed significantly increased cognitive functions including object exploration and recognition, passive avoidance, and spatial discrimination with significantly reduced levels of KYNA in the hippocampus [98]. Intriguingly, homozygous *got2*^−/−^ KO mice result in embryonal death in utero [99]. No study has been reported regarding KATs variants associated with psychiatric symptoms (Table 1).

### 3.4. Kynureninase

KYNU is a PLP-dependent hydrolase, encoded by the gene *kynu*, which catalyses L-KYN to AA and L-alanine and 3-HK to 3-hydroxyanthranilic acid (3-HAA) and (3-arylcarbonyl)-alanine. The enzyme also has Cys-conjugate-beta-lyase activity. KYNU is located in the cytosol [100]. However, there is a report that the enzyme has no activity for L-KYN and is inhibited by L-KYN and D-KYN. AA inhibits the TCA cycle and respiratory chain complexes I–III, interfering with mitochondrial function [101]. 3-HAA is also biosynthesized from AA by spontaneous hydroxylation. AA was once thought to be water-soluble vitamin L_1_ and is possibly related to an endogenous anti-inflammatory derivative in the celluar environment, as the molecule is a pharmaceutical precursor of nonsteroidal anti-inflammatory agents such as mefenamic acid and diclofenac [102]. 3-HAA can be either an oxidant or an antioxidant depending on the cellular condition. AA and 3-HAA suppress pro-inflammatory cytokines and invoke anti-inflammatory cytokine interleukin [IL]-10 [103].

*kynu*^−/−^ KO mice have been generated; however, no study has been reported regarding neurological and psychiatric diseases. Homozygous variant KYNU p.V57Efs*21 and heterozygous KYNU variants p.Y156* and p.F349Kfs*4 were identified in patients with vertebral, cardiac, renal, and limb defects syndrome 1, as well as autosomal recessive congenital malformation, characterized by vertebral segmentation abnormalities, congenital cardiac defects, renal defects, and distal mild limb defects. The NADH levels of patients are significantly lower [104]. KYNU SNP rs2304705 has been associated with essential hypertension and a 50% reduction in enzyme activity, but the activity reduction was not observed in another SNP [105].

In humans, increased levels of pro-inflammatory cytokine IL-1, associated with reduced levels of hippocampal neurogenesis, have been reported in depressed patients and in animal models of depression [106,107]. Identifying the mechanisms by which inflammatory cytokines block neurogenesis in the human brain may provide insight that can be used to manage inflammation-associated mental health disorders, including diet, new diagnostic methods, and treatment therapies for depression [108,109,110]. Recently, building on previous evidence, a new theoretical model named the neurovisceral integration model of fear (NVI-f), conceptualized the anatomical–functional interplay between the prefrontal cortex and heart-related dynamics in human emotional learning [111]. The potential of a novel approach, the time–frequency decomposition of heart rate variability, has gained attention in the evaluation of the abnormal fear learning that characterizes several neurological and psychiatric disoders [112]. Moreover, several studies have suggested the effectiveness of non-invasive brain simulation to interfere and modulate the abnormal activity of neural circuits, such as amygdala–medial prefrontal cortex (PFC) and hippocampus, involved in the acquisition and consolidation of emotional memories, which are altered in psychiatric disorders, such as fear-related disorder including anxiety disorder, phobias, post-traumatic stress disorder (PTSD), or depression [113,114,115,116] (Table 1).

### 3.5. 3-Hydroxyanthranilate 3,4-Dioxygenase, and toward the Tricyclic Carboxylic Cycle

3-hydroxyanthranilate oxidase (3-HAO) is the most active enzyme of Trp-KYN metabolic system, encoded by gene *haao*. The non-heme iron-dependent enzyme 3-HAO catalyzes 3-HAA to 2-amino-3-carboxymuconate semialdehyde (ACMS), which spontaneously cyclizes to QUIN. The enzyme is located in the cytosol [117]. QUIN forms a highly reactive hydroxyl free radical and is a NMDA receptor agonist which elicits excitotoxicity [118]. The concentration QUIN is increased upon immune activation and decreased by immune suppressant dexamethasone [119,120]. QUIN inhibits around 35% succinate dehydrogenase, an enzyme involved in the TCA cycle and in the respiratory chain [121]. *haao* gene variants HAAO p.D162* and HAAO p.W186* were identified in patients with vertebral, cardiac, renal, and limb defects syndrome 1, and the NADH level of patients was significantly lower [104].

Quinolinate phosphoribosyltransferase (QPRT) catalizes QUIN to nicotinic acid mononucleotide (NaMN), which is converted by NaMN adenyltransferase to nicotinic acid adenine dinucleotide (NaAD). Finally, NAD synthetase converts NaAD^+^ to NAD^+^. ACMS is catalized by 2-amino-3-carboxymuconate-6-semialdehyde decarboxylase (ACMSD) or picolinate carboxylase to 2-aminomuconic-6-semialdehyde (AMS) that is nonenzymatically cyclized to picolinic acid (PIC) [17]. The significantly lower levels of PIC were observed in ASD [122]. XA is converted to cinnabarinic acid (CA) by autoxidation. CA is also produced from 3-HK or QUIN. CA is an AHR agonist, and the reduced concentration of CA in the PFC is linked to SCZ [123]. 2-aminomuconate semialdehyde dehydrogenase (AMSD) catalyzes the conversion of AMS to 2-aminomuconic acid, which is further degraded to acetyl-CoA that replenishes the TCA cycle [102] (Table 1).

Currently, researchers are focusing on finding scientific frameworks for understanding the relationship between the molecular regulation of higher-order neural circuits and neuropathological alterations, and how this may lead to PFC dysfunction and to the symptoms of mental illnesses and comorbidity [124]. The deficit in control and motor inhibition [125,126], but also in motor imagery or in the suppression of on-going action [127], which depend on aberrant neural activity in the PFC associated with serious impulsivity problems, are characterized by psychopathological and psychiatric conditions including MDD, SCZ, obsessive–compulsive disorder (OCD), and PD [128,129].

Furthermore, the intentional neglect of adaptive procees necessary for memory functioning and functional alterations in the PFC affects the memory and learning abilities of psychiatric and brain-damaged patients. The human ventromedial PFC is responsible for the capacity of associative learning [130,131,132]. Hypoactivation in the ventromedial PFC with hyperactivation in the dorsal anterior cingulate cortex are reported in patients with PTSD and SCZ [133,134]. This evidence suggests that PFC dysfunctions cause impairment of aversive learning and emotional memory circuits, which might be transversal across many psychiatric disorders in humans as well as in neurologic patients [135].

## 4. Diseases Linked to Mitochondrial Dysfunction

The activity of neurons depends on mitochondrial function to elicit membrane excitability, execute neurotransmission, and maintain neuroplasticity [136]. Mitochondria are located throughout the cytoplasm of neurons, but more mitochondria are found in energy-demanding areas such as in the sites of branching axons, synaptic contacts, and glial processes [137]. The volume fraction of mitochondria is the highest in the cortical layer IV and mitochondrial volume is higher in dendrites than axons in rats, suggesting most energy consumption takes place at the postsynaptic side [138]. The dynamics of mitochondria are governed by mitochondrial fission, fusion, mitophagy, motility, and anchoring. Furthermore, mitochondria play a crucial role in axon degeneration and regeneration [139,140].

Mitochondrial dysfunction can affect any part of the body, but the most vulnerable organs are those with high-energy requirements, such as the central nervous system (CNS), peripheral nervous system, heart, and musculoskeletal system [141]. Mitochondrial diseases often go misdiagnosed or undiagnosed due to a wide range of manifestation, including fatigue; muscle weakness; visual or hearing loss; seizures; strokes; dementia; severe constipation; diabetes; thyroid or adrenal dysfunction; heart, liver, or kidney failure; poor growth; developmental delays; learning disabilities; and ASD in children [142]. The conditions can appear in adolescence and in adulthood [143]. The mitochondrial dysfunction also exhibits psychiatric symptoms such as depression, cognitive impairment, psychosis, and anxiety [144]. Currently, there is no cure for mitochondrial diseases and the mainstay of the treatment remains symptom-relieving or progression-delaying measures, which vary from patient to patient and depend on the mitochondrial disease and its severity [145]. Nevertheless, novel treatment for mitochondrial diseases is under extensive research. The strategies include oxidative stress modulation, mitochondrial biogenesis augmentation, mitochondrial autophagy modulation, nitric oxide restoration, mitochondria genome modulation, nucleotides pool restoration, hypoxia, enzyme replacement, and mitochondrial augmentation [146].

### 4.1. Primary Mitochondrial Diseases

PMDs are a clinically heterogenous group of uncurable, chronic, and genetic conditions caused by the mutations of mtDNA. PMDs commonly affect the nervous system of developmental stage, predominantly affecting skeletal muscles, but presenting many non-specific symptoms from muscle weakness to seizure [147]. The mutations of the genes may encode proteins functioning for OXPHOS, mtDNA replication and expression, mitochondrial dynamics, homeostasis, quality control, mitochondrial metabolism, metabolism of cofactors, and metabolism of toxic compounds, among others [148].

The severe forms of PMDs typically present early in life, but the milder forms tend to have later presentations [149]. Primary mitochondrial myopathy (PMM) causes progressive external ophthalmoplegia, frequently presented with diplopia, bilateral ptosis, or a head tilt. Progressive external ophthalmoplegia can be a part of a syndrome with facial muscle weakness or paralysis, swallowing difficulty, slurred speech, or breathing difficulty. Furthermore, PMM may show involvement of the muscles of the neck, shoulder, arms, hips, or legs, presenting cramping stiffness, weakness, pain, or paralysis of the affected muscles. Exercise intolerance is a common symptom [150]. Mitochondrial encephalomyopathy is characterized by neurological presentation in infancy or childhood, such as vision loss, sensorineural hearing loss, migraine, ataxia, or seizures. Other neurological manifestation includes dysphagia, dysarthria, myasthenia, or muscle rigidity. Some patients experience peripheral neuropathy. Developmental delays, failure to thrive, or short statue is a common finding in children [151]. In addition, many genetic disorders present mitochondrial myopathy or encephalomyopathy as a part of the main symptoms involved in multiple organ systems [152].

The diagnosis is made clinically but is very difficult and not always confirmed by a DNA mutation. The causative gene mutations can be located in both mtDNA and nDNA [153]. In total, 413 genes have been associated with PMDs. PMDs caused by mtDNA are estimated to have a prevalence of 1 in 5000 cases, while PMDs caused by nDNA are estimated to have prevalence of 1 in 35,000 [154]. The mutations can be either inherited or spontaneous. The mutations of nDNA can be inherited either autosomal dominantly or autosomal recessively. The mutations of mtDNA are considered to be inherited only maternally; however, a biparental mode of inheritance of mtDNA has been reported [155,156]. A variety of clinical manifestations in a single family may be due to the heteroplasmy of mtDNA or the different number of mutant mtDNAs in daughter cells as a result of mitosis.

Leigh syndrome is a neurodegenerative disorder, and it is the most prevalent mitochondrial disease in childhood. It is known that more than 75 genetical mutations appear in the basis of the disorder [157]. A recent study showed a reduction in the L-KYN and 3-HAA levels in blood with French Canadian variants of Leigh syndrome patients. In addition to this, the level of indoxyl sulfate increased in these patients which suggest a shift in Trp metabolism to the indol pathway [158]. Trp can metabolize not only into KYN or 5-HT, but also to indoxyl sulfate. Thus, Trp can transform to indole by tryptophanase, and indole metabolizes to indoxyl by cytochrome P450 2E1. Thereafter, sulfotransferase can convert indole to indoxyl sulfate [159].

Leber hereditary optic neuropathy is accompanied by a degeneration of retinal ganglion cells, causing a loss of vision [160]. The patients usually have some mutations in the genes, which encode complex I in the ETC. In patients with Leber hereditary optic neuropathy, decreased levels of Trp and glutamate have been found, suggesting the possible role of the Trp-KYN metabolic system in the pathomechanism of the disease [161,162].

### 4.2. Secondary Mitochondrial Dysfunction

SMD can be caused by genes not encoding proteins for OXPHOS or mitochondrial functions, secondary to other illnesses (such as cancer, sepsis, and infections, as well as metabolic, neuromuscular, neurodegenerative, and psychiatric diseases); by drugs such as tetracycline and valproate; by environmental factors (including alcohol, cigarette smoke, carbon monoxide, asbestos, and metals, as well as antiretroviral, tetracycline, valproate, and aminoglycosides therapy); or by normal aging [163,164,165]. Therefore, SMD can be inherited or acquired. The diagnosis of SMD is made based on clinical signs of mitochondrial dysfunction with negative or equivocal DNA tests. However, it is often difficult to distinguish SMD from PMD, but important for the prognosis and treatment. Sometimes, the treatment of PMD is effective to SMD [22].

### 4.3. Neurological Disesases Linked to Mitochondrial Dysfunction

The loss of stress resilience and the functional impairment of mitochondria have been linked to neuropsychiatric symptoms comorbidities of neurological diseases such as AD, PD, HD, ALS, FA, and Charcot–Marie–Tooth disease [166,167,168]. Preclinical animal research plays a major role in revealing the involvement of endogenous peptides, neurohormones, and metabolites including KYNs [169,170,171,172,173,174,175].

#### 4.3.1. Alzheimer’s Disease

AD is the most common chronic neurodegenerative disease with an insidious onset of progressive cytokine dysfunctions, particularly memory impairment, but it progresses to motor, sensory, and autonomic dysfunctions in later stages [176]. The age-related impairments in the ability to process contextual information and in the regulation of responses to threat are related to structural and physiological alterations in the PFC and medial temporal lobe [177]. Positron emission tomography with 2-deoxy-2-[fluorine-18]fluoro-D-glucose has shown the deficit of hubs in the theory of mind network in patients with mild cognitive impairment due to AD [178]. Brain autopsy and imaging studies reveal the atrophy of the brain including the frontal, temporal, parietal, entorhinal cortices, amygdala, and hippocampus [179]. The deposition of amyloid beta (Aβ) peptide and tau protein is a characteristic finding, but not limited to AD [180].

More than 170 genetically manipulated mouse models of AD have been created. Most transgenic mouse models of AD are designed to overexpress genes associated with early onset familial type of AD, such genes *APP*, *PSEN-1*, and *PSEN-2* genes, and the mouse strains are characterized with the pathological deposition of Aβ peptide [181]. Mitochondrial dysfunction including decreased mitochondrial respiration and pyruvate dehydrogenase protein was observed in the brains of triple transgenic mice of AD (3xTg-AD) at the age of 3 months, while mitochondrial Aβ levels significantly increased in 3xTg-AD at the age of 9 months. Mitochondrial impairment was even detected in embryonal neurons of 3xTg-AD [182]. Mitochondrial impairments were reported in a transgenic mouse expressing human amyloid precursor protein with the Arctic mutation (TgAPParc) mice at the age of six months. The mitochondrial membrane potential was decreased; the amount of reactive oxygen species was increased; and oxidative DNA damage was increased. Mitochondrial abnormality is more prominent in TgAPParc mice at the age of 24 months [183]. Mitochondrial dysfunction was also observed in transgenic mice carrying the APP_SWE_ and PSEN1_dE9_ mutations, heterozygous sodium-dependent vitamin C transporter (SVCT2^+/−^) KO mice, and transgenic APP/PSEN1 mice with heterozygous SVCT2 expression at the age of 4 months [184]. However, familial AD accounts for less than 5% of AD. Recently, human Aβ-knockin (KI) mice were engineered, which mimics a late-onset type. hAβ-KI mice develop age-dependent phenotypic and behavioral alterations and may be more relevant to study polygenic and multifactorial pathogenesis of AD [185].

The ratio of KYN/TRP was increased in the plasma and CSF of patients with AD and an increased 3-HK/KYN ratio in samples from CSF positively correlated with amounts of t-tau and p-tau peptides, while plasma KYN and PIC inversely correlated with p-tau and t-tau, respectively [186,187,188,189,190,191]. The levels of KYNA were found to be decreased in the plasma of AD patients [188]. AD had strong effect sizes for shared deficits in complex I and IV in the peripheral blood, frontal cortex, cerebellum, and substantia nigra [192] (Table 2).

#### 4.3.2. Parkinson’s Disease

PD is a progressive neurological disorder that affects the motor system with muscle rigidity, tremors, and changes in speech and gait. PD patients frequently experience non-motor symptoms (NMSs), such as sensory complaints, mental disorders, sleep disturbances, and autonomic dysfunction. NMSs often occur in PD due to the loss of dopamine-producing cells and the presence of Lewy bodies in the brain, having negative impacts on the quality of life and causing major challenges for disease management [193,194]. Human studies in healthy individuals have revealed that the modulation of autonomic nervous system responses is fundamental for behavioral regulation [195,196]. The pathogenesis of PD is considered to be largely due to the denervation of dopaminergic nigrostriatal nervous system and the aggregations of α-synuclein [194].

In a familial form of PD, mutations have been identified in genes encoding mitochondrion-associated proteins such as mitochondrial phosphatase and tensin homologue (PTEN)-induced kinase 1 (PINK1), Parkinson juvenile disease protein 2 (parkin), protein deglycase DJ-1 (Parkinson disease protein 7), and coiled-coil-helix-coiled-coil-helix domain containing 2 (CHCHD2) [197]. PINK1 KO mice may use a prodromal model of PD, as the mice show olfactory and gain disturbances [198]. The Parkin KO mouse is a classic transgenic PD model, while there are few studies using DJ-1 KO rats [199]. Homozygous CHCHD2 KO mice mimic PD pathology in an age-dependent manner; they are indistinguishable at birth, but fragmented mitochondria in dopaminergic neurons compared to the wild type [200]. NADH ubiquinone oxidoreductase core subunit S2 (NDUFS2) is a subunit of complex I in neurons that produce dopamine. The mitochondrial complex I-Park model that lacks the gene encoding NDUFS2 shows neurodegeneration [201]. However, transgenic mice that lack the gene encoding another complex I subunit NDUFS4 do not show neurodegeneration in dopaminergic neurons [202]. Complex I inhibitors 1-methyl-4-phenyl-1,2,3,6-tetrahydropyridine and rotenone are used for pharmacological models of PD, while 6-hydroxydopamine injections are applied for the oxidative stress model of PD [203]. Various types of α-syn transgenic mice do not develop significant nigrostriatal degeneration [204].

Significantly lower activities of KAT I and KAT II with a decreasing tendency of plasma KYNA levels were observed in the plasma samples of PD patients. Increased 3-HK levels in CSF and a lower KYNA/KYN ratio, increased QUIN levels, and higher QUIN/KYNA ratio were reported in plasma of PD patients [205,206]. In addition, it is reported that the single nuclear polymorphism variants of IDO1 influence the age onset of PD [207]. PD had strong effect sizes for shared deficits in complex I and IV in the peripheral blood, frontal cortex, cerebellum, and substantia nigra [192] (Table 2).

#### 4.3.3. Multiple Sclerosis

Multiple sclerosis (MS) is an autoimmune demyelinating neurodegenerative disease leading to the damage of neurons in the CNS. More common symptoms of MS range widely from motor and autonomic dysfunctions to psychobehavioral disturbances including pain, cognitive and emotional changes, and depression [208]. The neural lesion characteristic in MS is numerous plaques which are glial scars in the white matter and spinal cord [209]. The pathogenesis and progression of MS is ascribed at least partly to mitochondrial dysfunction, including reduced fidelity in gene expression, inadequate DNA repair, lower ATP supply, and increased reactive chemical species (RCS), among others [210]. Monitoring the redox status in patients with MS has been proposed to assess stress resilience and thus predictive biomarkers for therapeutic agents [211].

The most characterized animal models of MS are experimental autoimmune/allergic encephalomyelitis (EAE), Theiler’s murine encephalomyelitis virus-induced chronic demyelination, and toxin-induced demyelination [212,213]. After three days, immunization in EAE abnormal mitochondrial morphology appears such as vacuolization, swelling, and crista dissolution [214]. Focal intra-axonal mitochondrial alteration proceeds focal axonal degeneration, leading to axon fragmentation, which is triggered by macrophage-derived RCS [215]. Furthermore, the mitochondria of the spinal cord axon are depolarized, fragmented, and trafficking impaired in EAE mice [216]. Cuprizone produces cellular megamitochondria, leading to ATP shortage, RCS production, and ER stress in oligodendrocytes [217]. In addition, cuprizone mobilizes iron molecules from ferritin by chelating copper, leading to iron-mediated lipid peroxidation. The ferroptosis also leads to the production of more RCS via the Fenton reaction [218,219]. Upon withdrawal of the toxin treatment, the mitochondria reverse to normal original morphology [219,220].

Up to only a half of polymorphic loci is identified in the nuclear genome in MS inheritance. The rest of inheritable polymorphic variants may lie in the mitochondrial genome and interaction of mitochondrial and nuclear genes. The allele m.9055*G is found to be the mitochondrial variant associated with MS. The mitochondrial variants m.4216, m.4580, or m.13708 in biallelic combinations with nuclear gene variants of IL7R, CLEC16A, CD6, CD86, or PVT1 are found to be associated with MS [221]. Regarding KYNs, significantly decreased levels of KYNA were observed in the plasma and the brain tissue of mice treated with cuprizone [222].

The KYN/TRP ratio was significantly increased in the serum of MS patients. The QUIN levels were elevated, while NADH was decreased. 3-HK was found to be significantly higher in MS groups. The QUIN/KYNA ratio was higher in primary progressive MS, secondary progressive MS, and relapsing-remitting MS. KYNA levels were the highest in primary progressive MS, but lower in progressive MS [223]. The QUIN, neurofilament light, and neopterin levels were elevated in the CSF of MS patients compared to controls [224]. Significantly elevated QUIN/KYN and QUIN/KYNA ratios were observed in the CSF of the relapsing subgroup of relapsing-remitting MS. Trp, KYNA, and QUIN levels were increased in primary progressive MS, while Trp and KYNA levels were decreased in secondary progressive MS [225]. KYNA levels were significantly increased in the plasma of MS patients [226].

#### 4.3.4. Huntington’s Disease

HD is a fatal autosomal-dominant disease characterized by progressive and irreversible motor dysfunctions, leading to coordination and gait difficulties, as well as cognitive and behavioral changes. The degeneration and neural loss of the striatum, particularly the caudate nuclei, targeting the cerebral cortex, pallidum, thalamus, brainstem, and cerebellum, is a specific neuropathological finding in HD [227]. An abundance of ballooned neurons in the cerebellum, thalamus, and brain stem is another characteristic finding [180]. Mutant huntingtin (HTT) protein is associated with ballooning cell death.

The R6/1 and R6/2 mice are the first transgenic mice generated with a gene containing the promoter and exon 1 of human HTT with 115 or 150 CAT repeats, respectively. The transgenic mouse exhibits cognitive and motor deficits, irregular gait, clasping, weight loss, and seizure, resulting in early death [228]. The R6/2 model exhibited age-dependent changes in mitochondrial respiration in different regions of the brain [229]. The bacterial artificial chromosome transgenic mouse model of HD which carries full-length mutant HTT with a mixture of 97 CAG-CAA repeats exhibits progressive motor dysfunction, synaptic dysfunction, late-onset neuropathology, and neural degeneration [230]. Many KI models of HD have been generated [231]. The KI mice have the advantage of carrying a certain mutation under the endogenous *Hdh* promoter. Some models develop behavioral, molecular, cellular, and neuropathological phenotypes at an early age [231]. The role of aggregates remains unclear. The development of aggregates inhibitors has been under extensive research; however, the aggregates may not play an important role in the pathogenesis [232]. The homozygous HdhQ111 KI mutant huntingtin was found to be associated with the outer mitochondrial membrane, directly induced mitochondrial permeability transition (MPT) pore opening, and significantly decreased the Ca^2^^+^ threshold to trigger MPT pore opening [233]. KI mouse models exhibit the slow progression of behavioral abnormalities; thus, they may help reveal the pathomechanisms of HD and identify a new target for therapeutic intervention [234].

The lower levels of Trp and higher levels of KYN together with the higher KYN/Trp ratios were found in the serum of HD patients, suggesting the up-regulation of IDO activity [235]. The higher levels of 3-HK and QUIN and the higher activity of 3-HAO were observed in the striatum [236,237]. In contrast, the lower levels of KYNA and the lower activity of KATs were found in the plasma and the brain [238,239]. The levels of AA are found to be well correlated with the inflammatory status and the number of CAG repeats [240]. Thus, AA may be a potential prognostic biomarker for HD (Table 2).

#### 4.3.5. Amyotrophic Lateral Sclerosis

ALS is a progressive neurodegenerative disease causing the dysfunction of neurons controlling voluntary muscles. ALS often begins with fasciculation, myasthenia, or dysarthria, progressing to the involvement of the muscles responsible for moving, speaking, eating, and breathing [241]. Mitochondrial impairment is an early pathological event in ALS, leading to the death of motor neurons. The dysfunction of mitochondria affects calcium homeostasis, mitochondrial respiration, ATP production, mitochondrial dynamics, and apoptotic signaling. This is caused by the accumulation of ALS-associated mutant proteins such as superoxide dismutase 1 (SOD1), transactive response (TAR) DNA binding protein 43 kDa (TDP-43), fused sarcoma, chromosome 9 open reading frame 72 (C9orf72) gene product, and the C9orf72 GGGGCC repeat expansion-associated glycine/arginine dipeptide repeat protein [242].

Current rodent ALS models include the Friend leukemia virus B (FVB)-C9orf72 bacterial artificial chromosome (BAC) mouse that carries *C9orf72* mutations most associated with ALS, Cu/Zn SOD1-G93A mice that encode the human SOD1 protein containing the G93A mutation, and the TDP43-Q331K mouse model that mildly overexpresses human mutant TDP-43 [243]. The FVB-C9orf72 BAC mice develop paralysis and the loss of neuromuscular junction integrity, but the pathological manifestation depends on the mouse strain. The Cu/Zn SOD1-G93A mice show progressive motor dysfunction and loss of motor neurons, but it also depends on the mouse strain, and there is no evident upper motor neuron loss. The TDP43-Q331K mice develop progressive motor dysfunction with motor neuron and axon degeneration, but the progressive degeneration is limited in time, and it does not lead to death [243]. Morphological abnormalities have been observed in ALS models. Mitochondria were swollen in an induced pluripotent stem cell (iPSC) model of C9orf72-associated ALS [244]. The abnormal cluster formation of mitochondria was observed in the axon of SOD1 G93A transgenic mice [245]. Less elongated and more spherical mitochondria were isolated from the motor neuron of SOD1 G93A transgenic mice [246]. Expressions of wild-type or ALS TDP-43 mutants lead to abnormal morphology including aggregated, fragmented, and vacuolated mitochondria [247].

Environmental factors are considered to play a role in the pathogenesis of ALS and tabacco smoke has been linked to that of ALS. Exposure to bisphenol A (BPA) (a chemical used in the production of polycarbonate plastics) and beta-sitosterol beta-D-glucoside (BSSG) (an estrogen receptor-binding phytosterol) has been found to be neurotoxic to motor neurons; thus, those compounds have been applied to environmental models of ALS [243]. BPA induces neurotoxicity and neurodegeneration through alternations of mitochondrial functions, leading to fission and apoptosis via the translocation of dynamin-related protein 1 (Drp1) from the cytosol [248]. The neurotoxic effects of BSSG appear to be caused by mitochondrial production RCS associated with succinate oxidation (Complex II) [249].

Significantly increased TRP, KYN, and QUIN in serum and CSF, as well as significantly decreased PIC in serum, were observed in ALS. The neuronal and microglial expression of IDO and the levels of QUIN were increased in the motor cortex and spinal cord of ALS patients [250]. The levels of KYNA in ALS remain inconclusive, as studies showed significantly higher levels in the CSF, significantly lower levels in the serum, or no significant difference between healthy control and ALS patients [237]. Thus, the levels of KYNA may depend on the subgroup, the severity, and the stage of ALS, and further studies may reveal a potential role of KYNA measurement for biomarkers (Table 2).

#### 4.3.6. Migraine

Migraine is a primary headache disorder, characterized mostly by a headache on one side of the head. The exact pathomechanism of the disease is not fully known, but morphological and biochemical studies have shown that the pathophysiology of migraine is linked to mitochondrial dysfunction [251]. Abnormal mitochondria have been identified in patients with migraine with aura [252] and with familial hemiplegic migraine [253]. In addition, increased levels of lactate were shown in the blood and cerebrospinal fluid of patients with migraines, which clearly suggests a defective oxidative function [254,255]. The activities of mitochondrial enzymes including monoamine oxidase, succinate dehydrogenase, NADH dehydrogenase, cyclooxygenase, and citrate synthetase were found to be reduced in the platelets of migraineurs [256,257]. Furthermore, the biochemical changes were restricted to enzymes of the respiratory chain encoded by mtDNA [252]. OXPHOS has been found to be impaired in the brain of patients with migraines during and between migraine attacks [258,259,260]. This impairment is seen as increased levels of ADP, decreased levels of organic phosphate, and a decreased phosphorylation potential [261]. Reyngoudt et al. found that brain ATP decreased by 16% between attacks in patients with migraine without aura compared with healthy controls [262].

The mitochondrial involvements in migraine have been reported in animal models of migraine. The rodent inflammatory soup model revealed that mitochondria were small and fragmented, and that the number of mtDNA was significantly reduced in the trigeminal neurons. Furthermore, fission protein Drp1 was increased, while fusion protein mitofusin (Mfn) 1 was decreased, suggesting that mitochondrial dynamics were under disturbance after repeated dural stimulation [263]. The same chronic migraine model also showed that the trigeminal nucleus caudalis decreased spare respiratory capacity, i.e., the amount of ATP to be produced by oxidative phosphorylation in case of a sudden increased demand [264]. Neuroprotective, antiepileptic, and migraine prophylactic agent valproic acid stabilized the mtDNA copy number, restored the ATP level, and maintained the mitochondrial membrane potential in a rat model of nitroglycerin-induced trigeminovascular activation [265].

In the serum of chronic migraineurs, the levels of L-KYN, KYNA, 3-HK, 3-HAA, 5-HIAA, and QUIN were decreased, while the levels of L-Trp, AA, and XA were significantly higher compared to healthy controls [266]. The similar results were observed in patients with episodic or chronic cluster headache [267]. The levels of L-Trp, L-KYN, KYNA, 3-HAA, 5-hydroxyindolacetic acid, PIC, and melatonin were decreased in the plasma of episodic migraineurs in the interictal period. The tendency was more prominent in those without aura. In addition, the levels of 3-HAA, 5-hydroxyindoleacetic acid, and melatonin were increased in the ictal period [268]. The expression of KAT II was decreased in the upper cervical spinal cord (C1-C2) in nitroglycerin-induced trigeminovascular activation of rats [269]. Preclinical and clinical findings suggest that mitochondrial dysfunction and KYN metabolites play a role in the pathomechanism of migraine [270]. Furthermore, KYNs are also involved in the pathogenesis of chronic pain and their adjacent position to serotonin metabolism is drawing close attention to development of anti-migraine drugs [271,272,273,274]. The acute administration of antidepressant SSRI citalopram altered Trp-KYN metabolism in patients with migraines [275]. In addition, gastrointestinal disorders have been linked to migraines through Trp-KYN metabolism [276] (Table 2).

### 4.4. Psychiatric Disorders Linked to Mitochondrial Dysfunction

A growing number of researchers cast more attention on the contribution of mitochondria in mental health, susceptibility by genetic variants, and its interaction with environmental factors. Clinical and preclinical studies are revealing evidence that the organelles play a key role in psychiatric disorders and neurodevelopmental disorder such as MDD, generalized anxiety disorder (GAD), PTSD, SCZ, ADHD, and ASD. Furthermore, preclinical animal research plays a major role in revealing the involvement of endogenous neurotransmitters, neurohormones, and metabolites [277,278,279,280].

#### 4.4.1. Major Depressive Disorder

MDD is a mental disorder with at least two weeks of low mood, often accompanied by low self-esteem, loss of interest, low energy, and pain without a cause. The lifetime prevalence of MDD ranged from 2 to 21% [281]. The pathogenesis of depression has been linked to air pollution and depressive symptom frequently presents with cormobid conditions including anxiety, cognitive impairment, and chronic pain [282,283,284,285]. The monoamine hypothesis has prevailed for the pathogenesis of depression. The hypothesis holds that depression is caused by the depletion of 5-HT, norepinephrine, or dopamine in the CNS [286]. The atrophic lesion and synaptic impairment in the PFC and hippocampus and hypertrophy and increased synaptic activity in the nucleus accumbens, and amygdala are observed [287]. SSRIs are commonly used as a first-line treatment for MDD. However, only 42–53% of patients treated with SSRIs see an improvement, and medication for treatment-resistant depression remains a challenge. Novel treatment is under extensive study such as intermittent theta-burst stimulation [288,289]. Furthermore, psychotherapy is an effective treatment of choice, which can serve as a powerful measure for patients who cannot tolerate medication, and KYNs may be potentially useful as prognostic biomarkers [290].

A chronic mild stress (CMS) model of depression showed decreased ATP production, decreased hippocampal Na^+^ and K^+^-ATPase activity, and anhedonia in the sucrose preference test [291]. The damaged structure, impaired respiration rate, and altered membrane potentials of mitochondria were observed in the hippocampus, hypothalamus, and the cortex of CMS mice which exhibit anhedonia in the sucrose preference test and depression-like behavior in the tail suspension test (TST) [292]. Furthermore, SSRI fluoxetine demonstrated Na^+^ and K^+^-ATPase activity, mitochondrial respiration, and sucrose preference in the chronic unpredictable stress model [293]. Thus, mitochondrial dysfunction may be involved in depression-like behavior.

Transgenic models for depression have been generated by manipulating genes responsible for the metabolism of 5-HT. Tryptophan hydroxylase (TPH) is the rate-limiting enzyme in 5-HT biosynthesis. The *Tph1*^−/−^ mice produced a normal level of 5-HT in the brain and showed no significant change in behavior [294,295]. The *Tph2*^−/−^ mice exhibited depressive-like behavior in TST and anxiety-related behavior in the marble burying test [296]. However, it was reported that Tph2 null mutants (*Tph2*^−/−^) mice showed slightly reduced depression-like and anxiety-like behaviors, but significantly increased fear-conditioning responses. The behaviors, including impulsivity, aggressiveness, and emotional reactivity of *Tph2*^−/−^ mice, are sex-dependent [297]. The double KO *Tph1/Tph2*^−/−^mice showed depressive-like behavior in TST and anxiety-related behavior in the marble burying test but antidepressive-like behavior in the forced swim test (FST) with reduced levels of 5-HT in the brain and periphery [298]. In addition, KI of the TPH2 variant (R439H) in mice showed depression-like behavior in TST [299].

MDD patients showed moderate effect sizes for similar abnormality patterns in the expression of complex I of samples from frontal cortex, cerebellum, and striatum [192]. The levels of Trp, KYN, and KYNA were decreased in the plasma of MDD patients, and the levels of QUIN were increased in MDD patients without antidepressant treatment. The immunoreactivity of QUIN was increased in the PFC and hippocampus of the postmortem brain tissues from MDD patients [300,301]. An increased risk of depression was reported following the activation of the Trp-KYN metabolic system in chronic illnesses [302]. Furthermore, KYNA may serve as a diagnostic and predictive biomarker in depression [303]. The serum KYNs were found to be correlated with depression in poststroke patients [304]. KYN analogues have been extensively researched in the search for novel antidepressants [305] (Table 3).

#### 4.4.2. Generalized Anxiety Disorder

GAD is a mental disorder characterized by excessive, uncontrollable, and irrational anxiety. GAD has a combined lifetime prevalence of 3.7% [306]. 5-HT, dopamine, norepinephrine, and gamma-aminobutyric acid (GABA) are linked to anxiety [307]. The amygdala in the middle of the brain which processes emotion, memory, and fear is involved in GAD [308]. Benzodiazepines such as alprazolam, clonazepam, and diazepam bring relief in 30 min; SSRIs are the first line of treatment in GAD; and cognitive behavioral therapy is the most effective form of psychotherapy [309].

An increasing number of preclinical studies are revealing that anxiety is linked to mitochondrial functions including bioenergetics, oxidative stress, neurosteroid production, biogenesis, and apoptosis [310]. Mitochondrial dysfunction is located in the nucleus accumbens (NAc) which interfaces motivation and action, playing a key role in motivation, aversion, reward, and reinforcement learning. The shell of NAc is considered to be part of the extended amygdala [310]. The outbred Wistar rats which exhibit anxiety-like behavior showed reduced expressions of the mitochondrial GTPase MFN 2 in the NAc, altered mitochondrial morphology and functions, and the morphology of medium spiny neurons (MSNs) projecting from the NAc. The behavioral, mitochondrial, and neuronal phenotypes were reversed by the viral overexpression of MFN2 [311]. Furthermore, more anxious rats are prone to become subordinate during a social encounter with less anxious rats and social hierarchy is linked to the mitochondrial bioenergetic profiles of the NAc. Thus, anxiety appears to directly influence social dominance mediated by mitochondrial functions [312].

The levels of KYN were reduced in endogenous anxiety and normalized after treatment in the plasma samples [313]. The levels of KYN were lower in people with Type D personality, the joint tendency towards negative affectivity and social inhibition [314].

#### 4.4.3. Post-Traumatic Stress Disorder

PTSD is a behavioral and mental disorder that develops after experiencing a traumatic event. Individuals with PTSD suffer from flashbacks, nightmares, severe anxiety, and uncontrollable thoughts regarding the event [315]. The lifetime prevalence of PTSD ranges from 6.1 to 9.2% [316]. PTSD is considered to be caused by insufficient integration of a trauma memory into the hippocampal-cortical memory networks, forming fragmented, incomplete, and disorganized intrusive memories [317]. The primary treatment of PTSD is psychotherapy, and SSRIs such as sertraline and paroxetine are considered first-line therapy for PTSD [318].

Animal models are revealing the pathogenesis of PTSD. The genetic factors contributing to the development of PTSD include the stress response system, such as the hypothalamic–pituitary–adrenal (HPA) axis; neuroplasticity such as brain-derived neurotrophic factor (BDNF); and monoamine neurotransmission such as serotonergic, dopaminergic, glutamatergic, and GABA-ergic systems [319]. FK506-binding protein 51 (FKBP5) is a co-chaperone which modulates glucocorticoid receptor activity. FKBP5^−/−^ mice prevents the age-induced impairment of stress resilience [320]. Chronic variate stress increases the bed nucleus of the stria terminalis pituitary adenylate cyclase activating polypeptide (PACAP) [321]. The pituitary adenylate cyclase 1 receptor type 1 KO (PAC1R^−/−^) mice show reduced anxiety [322]. The BDNF promotor IV-disrupted mutant Bdnf-e4 mice and BDNF Met-Val mutant mice showed impaired fear extinction [323,324].

Serotonin 1A receptor KO 5-HT1AR^−/−^ mice show increased fear memory to contextual cues [325]. 5-HT transporter (5-HTT) gene KO 5-HTT^−/−^ mice show impaired stress response and impaired fear extinction with abnormal corticolimbic structure [326]. Dopamine is degraded by catechol-*O*-methyltransferase (COMT). COMT gene KO COMT^−/−^ mice showed an increased response to repeated stress exposures [327]. GABA is synthesized from L-glutamic acid by glutamic acid decarboxylase. The 65-kDa isozyme of glutamic acid decarboxylase 2 KO GAD6^−/−^ mice shows increased generalized fear and impaired extinction of cued fear [328,329]. GABA receptor subunit B1a KO GABAB1a^−/−^ mice shows a generalization of conditioned fear to nonconditioned stimuli [330]. Cannabinoids directly interact with GABAergic neurotransmission. Cannabinoid 1 receptor KO CB1R^−/−^ mice show an increased response to repeated stress exposures [331].

Mitochondrial functions are linked to PTSD-like behavior in preclinical studies. Following exposure to a trauma, mice with PTSD-like symptoms exhibit reduced activities of mitochondrial electron transport in the cerebellum and the dysfunction of fatty acid oxidation in cerebellum and plasma. The activity of cerebellar mitochondrial electron transport complex is negatively correlated with PTSD-like symptoms [332]. Abnormal apoptosis has been observed in the brain areas closely associated with emotion and cognition, including the hippocampus, the amygdala, and the medial PFC in single prolonged stress model of PTSD [333]. In addition, the decreased kynurenine pathway potentiates resilience to the social defeat effect of a cocaine reward [334]. Early intervention with a glucocorticoid receptor antagonist a RU486 facilitates the correction of traumatic stress-induced fear and anxiety dysregulation [335]. No clinical study was reported regarding the peripheral or CSF samples of KYNs in patients with PTSD. KYN metabolites are monitored in clinical settings as evidence of inflammatory responses contributing to sleep deprivation and the formation of intrusive memories [336] (Table 3).

#### 4.4.4. Bipolar Disorder

BD is a mental disorder characterized by mood oscillations with episodes of mania and depression. A large cross-sectional survey of 11 countries found the overall lifetime prevalence of BD was 2.4% [337]. Neuroimaging and postmortem studies have found abnormalities in a variety of brain regions, and the most commonly implicated regions include the ventral PFC and the amygdala [338,339]. Dysfunctions in emotional circuits located in these regions have been hypothesized as a mechanism for BD. The left side of the hippocampus regulates verbal and visual memory. This part of the brain also helps regulate how you emotionally respond to situations. When your mood shifts, your hippocampus changes shapes and shrinks [340]. Patients with BP showed diminished GABA neurotransmission. Thus, low GABA levels can result in excitatory toxicity [341]. 

There have been no established animal models of BD that exhibit both manic and depressive episodes. Typical current animal models of mania involve drug-induced hyperactivity or genetically modified animals that exhibit continuous hyperactivity. The targeting of circadian rhythm genes to disrupt mechanisms regulating the circadian rhythm has been widely used to create animal models for BD [342].

The most common model is the *ClockΔ19* mutant mouse. These mice carry a deletion at exon 19 of the *Clock* gene, resulting in a dominant-negative protein, unable to activate transcription [343]. Mutant mice exhibit mania-like behavior and altered sleep patterns [344]. The dominant negative mutant of mtDNA *Polg1* transgenic mice showed recurrent hypoactive periods [345]. The withdrawal of lithium provokes depression in mice, while antidepressant medications alleviate depressive symptoms [346]. Thus, the transgenic strain appears to be a good animal model for BD.

Meta-analysis revealed that BD showed moderate effect sizes for similar abnormality patterns in the expression of complex I of samples from frontal cortex, cerebellum, and striatum [192]. The dysfunctional mitochondrial hypothesis is one of the current hypotheses that attempt to explain the origin of mood disorders. Many studies have confirmed that mood stabilizers affect mitochondrial functions, even though the exact mechanism or localization of action is unknown [347].

Regarding the KYN system a case-control study showed that KYNA levels were reduced and the 3-HK/KYN and 3-HK/KYNA ratio was increased in BD compared to healthy control [348]. However, a meta-analysis reported no significant difference of TRP and KYN levels, KYN/TRP and KYNA/QUIN ratios in serum from BD patients [349] KYNA was significantly increased in CSF of BD patients [350] (Table 3).

#### 4.4.5. Substance Use Disorders

Substance use disorders (SUDs) represent a type of mental disorder that affects the brain and behavior, leading to an inability to control the use of a drug or medication. The exact cause of SUDs is not known, but the known risk factors are the genes, the action of the drug, peer pressure, emotional distress, anxiety, depression, and environmental stress [351]. In addition to an impaired control, common substances are alcohol, sedatives, caffeine, hallucinogens, inhalants, stimulants, and tobacco, among others [352]. The main brain area associated with SUDs is the limbic system, comprising the cingulate gyrus, amygdala, hippocampus, PFC, ventral tegmental area, and the nucleus accumbens. The system is related to reward, emotion, and punishment [353]. 

The mitochondrial copy numbers were found to be reduced in blood samples of patients with opioid use disorder; however, the link between changes in the reward neural circuitry and the peripheral measurements remains unclear [354]. No clinical study was found regarding KYNs in patients with SUDs. Although there are few studies on this, growing attention is paid to a relationship between KYN metabolites and SUDs, the alteration of the Trp-KYN system, and a potential approach to SUDs (including ethanol, nicotine, cannabis, amphetamines, cocaine, and opioids) [355]. Furthermore, 5-HT concentration was significantly higher and the KYN/5-HT ratio was significantly lower in plasma of patients with cocaine use disorder in SUD-induced MDD compared to those with MDD, but there were no differences between SUD primary MDD and MDD. This may suggest that the Trp-KYN pathway participates less in SUD-induced MDD [356].

#### 4.4.6. Schizophrenia

SCZ is a mental disorder characterized by abnormally interpret reality, hallucinations, delusions, apathy, lack of social functioning, and extremely disordered thinking and behavior. Cognitive symptoms, including concentration and attention difficulties, as well as memory impairments, can be subtle [357]. Cognitive deterioration in patients with SCZ has been linked to vitamin D deficiency which may directly affect processing speed [358]. SCZ is generally considered to be a neurodegenerative disorder with neurodevelopmental antecedents. The underlying changes occur before the onset of symptoms arising from the interaction between genes and the environment, leading to deficits in the neural circuitry in the age of 18–25 [359]. Maternal infections, malnutrition, and complication during pregnancy and parturition are risk factors [360]. In total, 30–50% of SCZ patients develop antipsychotic-resistant SCZ which is associated with a high level of dissociation, a loss of integrity between memories, and perceptions of reality [361]. Many people with SCZ have hypertension, disturbance of lipid metabolism, and other mental disorders (including SUDs, MDD, GAD, and OCD) [362,363,364].

Disrupted in schizophrenia 1 (DISC1), encoded by the *DISC1* gene, is a protein which plays a role in presynaptic regulation of dopamine. DISC1 alterations increase the risk of SCZ [365]. DISC1 also plays various roles in many other cellular functions, including mitochondrial transport, fission, and fusion. The dynamic processes of mitochondrial transport, fission, and fusion determine mitochondrial morphology, localization, and network [366]. DISC1 mouse models display abnormal changes relevant to SCZ. The neuroanatomical changes include displaced dentate granule neurons, altered axonal targeting, reduced dendrite growth, and dendritic spine density. The behavioral abnormalities include the impairment of working memory [367]. Furthermore, the spontaneously hypertensive rat strain haves been proposed to an animal model of SCZ, which exhibits abnormal behaviors resembling cognitive, psychotic, and negative symptoms [368].Neurodegenerative changes in SCZ are caused by a series of malfunction including mitochondrial impairment, oxidative stress responses, and the activation of immune responses, leading to chronic low-grade inflammation [17]. Clinical studies linked mitochondrial impairment with increased risk of SCZ and suggested that abnormal mitochondrial dynamics contribute to compromising normal neural connectivity in the brain [369,370]. Furthermore, meta-analysis revealed that SCZ showed moderate effect sizes for similar abnormality patterns in the expression of complex I in samples from the frontal cortex, cerebellum, and striatum [192].

Regarding the Trp-KYN metabolic system, KYN and the ratios of KYN/TRP were higher in the serum of SCZ patients [371]. Meta-analyses showed increased KYN and KYNA levels in CSF samples of SCZ patients and increased levels of KYNA in plasma, CSF, brain tissue, or saliva, respectively [53,256]. Thus, the KYN system is activated in SCZ, and elevated KYNA levels are considered to contribute to the impairment of cognitive function. However, another meta-analysis reported that KYNA levels and the KYNA/3-HK ratio were not altered, and the KYNA/KYN ratio was decreased in SCZ, suggesting the presence of differential patterns between SCZ and mood disorders [53] (Table 3).

#### 4.4.7. Autism Spectrum Disorder

ASD is a neurodevelopmental disorder characterized by persistent deficits in social interaction, restricted-repetitive patterns of behavior, and the loss of interests or activities [372]. These social impairments may be related to the interpretation of social signals [373]. Potentially threatening situations, such as the proximity of others, can trigger a number of physiological responses that help to regulate the distance between themselves and others during social interaction, showing the critical role of social signal interpretation in social interaction. Individuals with ASD have social impairments, potentially due to the lack of social signal interpretation, resulting in an inability to interpret these signals to guide appropriate behaviors. The prevalence of mitochondrial diseases is higher in the population of ASD than in general population and up to a half of children with ASD showed evidence of mitochondrial dysfunction [374]. However, most mitochondrial disease-associated ASD is not associated with genetic abnormalities, suggesting secondary mitochondrial impairment, such as environmental factors [375].

Animal models of ASD include prenatal exposure to valproate during pregnancy, inbred strains of mice expressing autism traits, and genetical modification targeting autism-related genes (including mtDNA) [376]. The insertion of mtDNA *ND6* gene missense mutation (*ND6^P25L^*) exhibits ASD endophenotypes, including autism-like behaviors and electroencephalographic profiles, and correlates with mitochondrial respiration and increased RCS of the brain, suggesting a link to mitochondrial dysfunction [377]. The hemi-deletion of the Src-homology 3 and the multiple ankyrin repeat domain 3 (SHANK3) gene is found in patients with Phelan–McDermid syndrome, demonstrating ASD-like behaviors. The homozygous Shank3^Δc/Δc^ mice with C-terminal 508 deletions exhibit significant impairments in social novelty preference, stereotyped behavior, and gait [377]. The contactin-associated protein-like 2 (*CNTNAP2)* encodes a neurexin that regulates the interactions of neurons and glial cells. The *Cntnap2* knockout mice show ASD-like behavior [378]. The adhesion of G protein-coupled receptor L3 (*ADGRL3*) encodes latrophilins. *ADGRL3*^−/−^ mice show hyperactivity and less depression-like behavior. Preclinical studies also showed links between the pathogenesis of ASD and maternal immune activation, maternal microbiota profile, and exposure to nutritional and toxic metals during mid-fatal development [379,380,381,382].

The alteration of the Trp-KYN metabolic system was also observed in patients with ASD. The levels of KYNA were significantly lower, and the ratio of KYN/KYNA was significantly higher in the serum of children with ASD [383]. The ratio of KYN/Trp and the levels of KYN and QUIN were significantly higher in blood samples of ASD patients, but there was no significant difference in KYNA and the levels of PIC were significantly lower in ASD patients [121] (Table 3).

#### 4.4.8. Attention-Deficit Hyperactivity Disorder

ADHD is a behavioral and neurodevelopmental disorder characterized by inattention, hyperactivity, and impulsivity, which are pervasive, impairing, and otherwise age-inappropriate [384]. ADHD is associated with SUDs, alcoholism, and other mental disorders, including MDD, GAD, and ASD [385]. Furthermore, multidirectional relationships between stress, anxiety, and inflammation in the pathogenesis of ADHD are discussed recently [386].

Patched domain-containing protein (*Ptchd*) is a membrane protein with a patched domain. The deletion of the *Ptchd* gene has been identified in patients with intellectual disability and ASD. The *Ptchd1* KO mice exhibit ADHD-like behaviors. No changes in Trp and 3-HK were found, but significant increased levels of KYN, KYNA, AA, and 3-HK were observed in the serum of the *Ptchd1* KO mice. Meanwhile, significantly increased levels of AA, 3-HK, and 3-HAA were observed in the frontal cortex of *Ptchd1* KO mice, but there were no changes in KYNA levels [387]. A clinical study showed that lower concentrations of Trp, KYNA, and XA, 3-HAA were found in the serum of patients with ADHD, and that higher levels of Trp and KYN were associated with higher scores of ADHD symptoms [388].

Mitochondria may be sensitive to psychological stress in early life [389]. People who experienced childhood trauma appear to possess a larger number of mitochondrial genomes per cell [389]. Indeed, mtDNA copy number was observed to be higher in the peripheral blood of ADHD patients, which suggests a possible link to mitochondrial impairments in the pathogenesis of ADHD [390] (Table 3).

## 5. Conclusions and Future Perspective

This review article recapitulated the involvement of mitochondria with an emphasis on its connection to the Trp-KYN metabolic system in clinical manifestations of neuropsychiatric symptoms and advances in preclinical research in major neurological and psychiatric disorders. Growing evidence has revealed that mitochondria have a close link to KYN metabolism and that mitochondrial dysfunction and the activation of the KYN system contribute to the pathogenesis of neuropsychiatric disorders. Extensive clinical and preclinical research has helped delineate the multifunctional facets, compartmentalization, and dynamic nature of mitochondria, including cell differentiation, cell-type determination, cell movement, and pattern formation. The pathological changes in functions, morphologies, and dynamics have been probed in mitochondrial diseases, as well as diseases linked to mitochondrial dysfunction. For example, functional magnetic resonance imaging, measurements of fibroblast mitochondrial spare respiratory capacity, the NAD^+^/NADH ratio, Complex II levels, and a combination of the detection of amyloid and/or tau protein and signs of neuronal injury on brain imaging or cerebrospinal fluid sampling are emerging techniques used to assess mitochondrial functions.

However, little is known about the reversibility, plasticity, and/or resilience of mitochondrial functions, integrity, dynamics, and/or network formation. The measurement of such parameters is of particular importance. Revealing the link between mitochondria and the KYN metabolic system may be a promising option for this direction of research. The development of the mitochondrial stress test, for example, which assesses recoverability, may help with the early detection of mitochondria-related diseases and the possible application of prophylactic measures. For this purpose, engineering fine preclinical models high in construct, face, and predictive validity is an essential step. Two-hit models consisting of a certain genetic susceptibility and environmental trigger with pharmacological agents which initiate neuropsychiatric manifestations can help develop preventive measures, understand the pathomechanism, make accurate diagnoses, delay disease progression, and choose the most appropriate therapeutic option.

## Figures and Tables

**Figure 1 cells-11-02607-f001:**
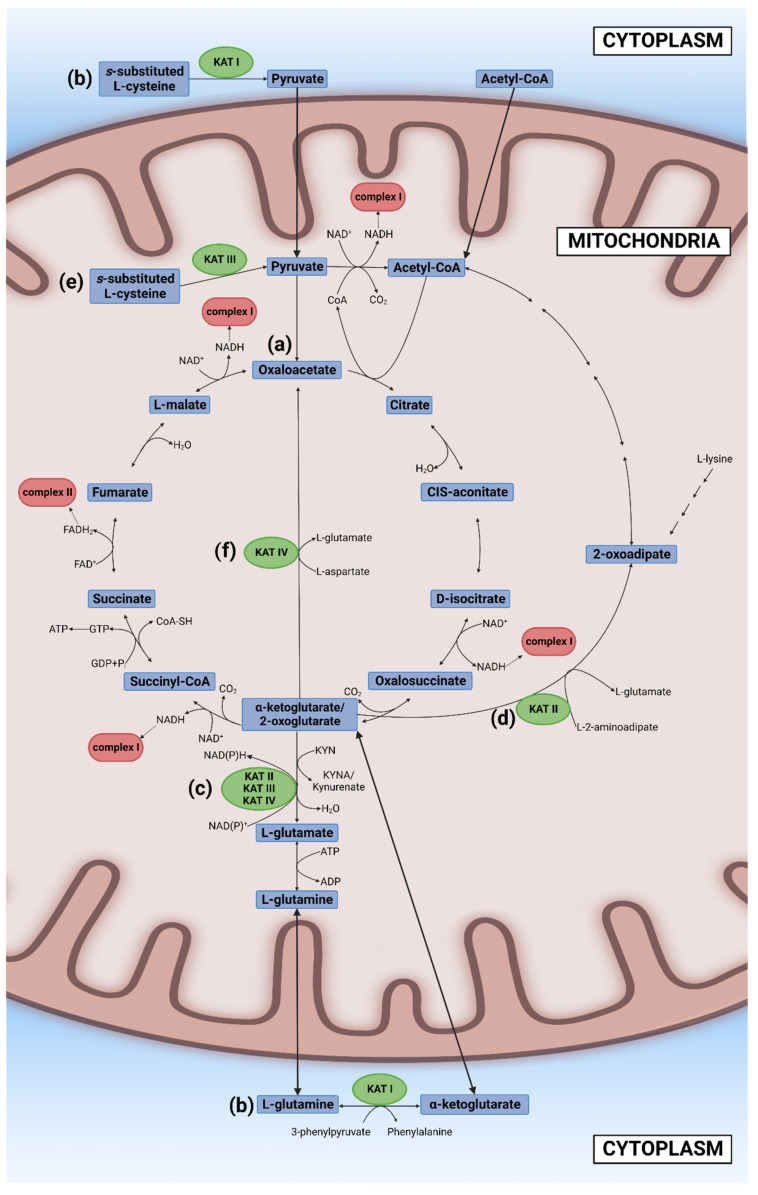
The tricarboxylic acid cycle (TCA) and its interface with the tryptophan (Trp)–kynurenine (KYN) metabolic system. (**a**) The TCAcycle is initiated with acetyl coenzyme A (acetyl-CoA) reacting with oxaloacetate to form citrate. Citrate is oxidized to alpha (α)-ketoglutarate (2-oxoglutarate) with the formation of nicotinamide adenine dinucleotide (NADH). α-ketoglutarate is oxidized to succinyl coenzyme A (succinyl-CoA) with the formation of NADH. Succinyl-CoA is converted to succinate with the formation of adenosine triphosphate (ATP). Succinate is oxidized to fumarate with the formation of flavin adenine dinucleotide (FADH_2_). Fumarate is hydrated to malate which is oxidized to oxaloacetate to end the cycle. (**b**) Cytosolic kynurenine aminotransferase (KAT) I catalyzes the reaction of an *S*-substituted L-Cys to pyruvate. KAT I also catalyzes the reaction of L-glutamine to α-ketoglutarate (2-oxoglutarate). (**c**) Mitochondrial KAT II, KAT III, and KAT IV catalyze the reaction of α-ketoglutarate catalyzes the reaction of L-glutamine to α-ketoglutarate (2-oxoglutarate) to L-glutamate. (**d**) KAT II catalyzes the reaction of α-ketoglutarate (2-oxoglutarate) to 2-oxoadipate which is eventually degraded to acetyl-CoA. (**e**) Mitochondrial KAT III catalyzes the reaction of an *S*-substituted L-Cys to pyruvate. (**f**) Mitochondrial KAT IV catalyzes the reaction of α-ketoglutarate (2-oxoglutarate) and L-aspartate to α-ketoglutarate (2-oxaloacetate) and L-glutamate.

**Figure 2 cells-11-02607-f002:**
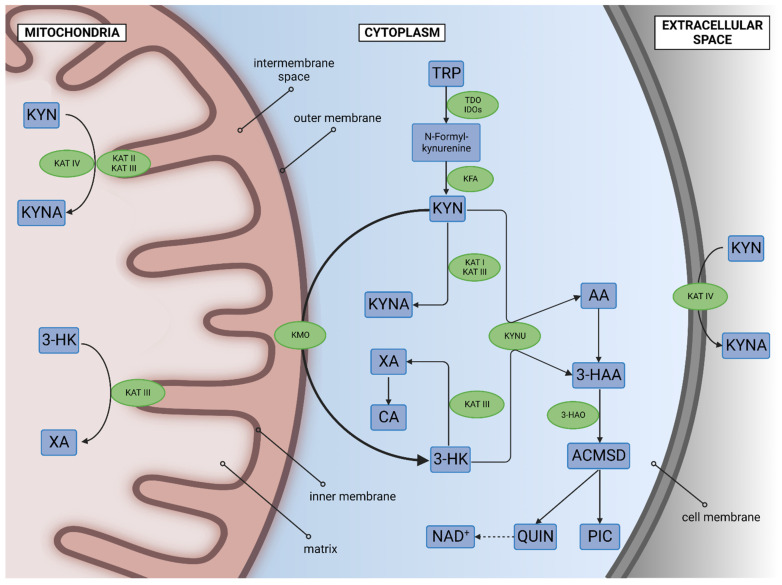
The tryptophan (Trp)–kynurenine (KYN) metabolic system and the subcellular location of the enzymes. More than 95% of L-Trp enters the KYN system producing multifarious biomolecules. The main enzymes of the KYN system are tryptophan 2,3-dioxygenase (TDO), indoleamine 2,3-dioxygenases (IDOs), kynurenine formamidase (KFA), kynurenine 3-monooxygenase (KMO), kynurenine aminotransferases (KATs), and kynureninase (KYNU). Most of the enzymes are located in the cytosol. However, KMO is located in the outer membrane of mitochondria; KAT II and KAT III are in the inner membrane of mitochondria; and KAT IV is in the matrix of mitochondria and in the plasma membrane. The main metabolites are L-KYN, kynurenic acid (KYNA), 3-hydroxy-L-kynurenine (3-HK), quinolinic acid (QUIN), and nicotinamide adenine dinucleotide (NAD^+^) which exhibit a wide range of biological properties and the metabolites freely cross the mitochondrial membranes. AA: anthranilic acid; ACMSD: amino-β-carboxymuconate-semialdehyde-decarboxylase; CA: cinnabarinic acid, 3-HAA: 3-Hydroxyanthranilic acid, 3-HA0: 3-hydroxyanthranilate oxidase; PIC: picolinic acid; XA: xanthurenic acid.

**Table 1 cells-11-02607-t001:** The enzymes, genes, substrates, products, activities, knockout, and human gene variants of the tryptophan–kynurenine metabolic system.

Enzymes	Genes	Substrates	Products	Locations	Transgenic Models	Animal Traits	Human Gene Variants
TDO	*tdo2*	L-Trp	N-formyl-L-kynurenine	Cytosol	*tdo2* ^−/−^	- Anxiety (inconclusive) - Exploratory activities (inconclusive) - Cognitive function	- ADHD, MDD, ASD, SCZ, Tourette syndrome
IDO1	*ido1*	L-Trp, D-Trp	N-formyl-L-kynurenine	Cytosol	*ido1* ^−/−^	- Locomotion - Nociception - Depression	- MDD
IDO2	*ido2*	L-Trp	N-formyl-L-kynurenine	Cytosol	*Ido2* ^−/−^	-	- MDD
KFA	*afmid*	N-formyl-L-kynurenine	L-KYN	Cytosol	-	-	-
KMO	*kmo*	L-KYN	3-HK	Mitochondria (outer membrane)	*kmo* ^−/−^	- Lower contextual memory function - more anxious-like behavior - Higher horizontal activity upon a D-amphetamine challenge	- Cognitive dysfunction - Lower cognitive performance - A trend effect on cognitive function
KAT I (Kynurenine--oxoglutarate transaminase 1)	*kyat1*	L-KYN S-substituted L-Cys 3-phenylpyruvate L-glutamine	KYNA thiol, NH_4_, pyruvate 2-oxoglutarate L-phenylalanine	Cytosol	-	-	-
KAT II (kynurenine/α-aminoadipate aminotransferase) (KAT/AadAT)	*aadat*	L-KYN α-ketoglutarate 2-oxoglutarate	KYNA L-glutamate L-glutamate	Inner membrane of mitochondria	*aadat*^−/−^ (aka *kat2*^−/−^)	- Transitory hyperlocomotive activity - Transitory abnormal motor coordination - Increased cognitive functions	-
KAT III (kynurenine--oxoglutarate transaminase 3)	*kyat3*	L-KYN α-ketoglutarate 3-HK glyoxylate glyoxylate L-KYN *S*-substituted L-Cys H_2_O	KYNA L-glutamate Glycine H_2_O XA Glycine H_2_O KYNA Thiol NH_4_^+^ pyruvate	Cytosol inner membrane of mitochondria	-	-	-
KAT IV (aspartate aminotransferase, mAspAT)	*got2*	L-KYN α-ketoglutarate 2-oxoglutarate L-aspartate	KYNA L-glutamate L-glutamate oxaloacetate	Matrix of mitochondria plasma membrane	-	-	-
KYNU	*kynu*	L-KYN L-alanine 3-HK	AA 3-HAA (3-arylcarbonyl)-alanine	Cytosol	*kynu* ^−/−^	-	- Vertebral, cardiac, renal, and limb defects syndrome 1 - Essential hypertension
3-HAO	*haao*	3-HAA	ACMS	Cytosol	-	-	- Vertebral, cardiac, renal, and limb defects syndrome 1

**Table 2 cells-11-02607-t002:** The preclinical models, mitochondrial involvement, and findings in kynurenines in main neurological diseases.

Neurological Diseases	Preclinical Models	Mitochondrial Involvement	Findings in Kynurenines
Alzheimer’s disease	>170 genetic models (APP, PSEN-1, PSEN-2)	-	- increased ratio of KYN/Trp - decreased KYNA - 3-HK/KYN positively correlated with t-tau and p-tau peptides - KYN and PIC negatively correlated with t-tau and p-tau peptides
	3xTg-AD	- decreased mitochondrial respiration - decreased pyruvate dehydrogenase protein - increased mitochondrial Aβ level	
	TgAPParc	- decreased mitochondrial membrane potential - increased reactive oxygen species - increased oxidative DNA damage - mitochondria impairments	
	APP_SWE_	-	
	PSEN1_dE9_	-	
	SVCT2^+/−^	-	
	human Aβ-KI	-	
Parkinson’s disease	PINK1 Parkin Parkinson disease protein 7	- - -	- lower activities of KAT I and KAT II - decreased KYNA - increased 3-HK - lower KYNA/KYN ratio - increased QUIN - higher QUIN/KYNA ratio
	CHCHD2	- fragmented mitochondria	
	complex I Park model	- neurodegeneration	
	methyl-4-phenyl-1,2,3,6-tetrahydropyridine	-	
	Rotenone	-	
	6-hydroxydopamine	-	
Multiple sclerosis	experimental autoimmune/allergic encephalomyelitis (EAE)	- depolarized fragmented mitochondria - trafficking-impaired	- increased KYN/TRP ratio - decreased NADH - higher 3-HK - higher QUIN/KYNA ratio - Trp, QUIN, KYNA depending on subtypes - higher QUIN - higher QUIN/KYN ratio
	Theiler’s murine encephalomyelitis virus-induced chronic demyelination	-	
	cuprizone-induced demyelination	- megamitochondria	
Huntington’s disease	R6/1	-	- lower Trp - higher KYN - higher KYN/Trp ratio - higher 3-HK - higher HAO activity - lower KYNA - lower KAT activity - AA levels correlated with the number of CAG repeats
	R6/2	-	
	HTT+97CAG-CAA repeats	-	
	KI (endogenous Hdh promoter)	-	
	HdhQ111KI	- multiple mitochondria abnormality	
Amyotrophic lateral sclerosis	FVB-C9orf72 BAC	-	- increased TRP, KYN, QUIN - decreased PIC - KYNA inconclusive
	Cu/Zn SOD1-G93A	-	
	TDP43-Q331K	-	
	iPSC model of C9orf72-associated ALS	- swollen mitochondria - cluster formation of mitochondria - elongated spherical mitochondria - mitochondrial fission and apoptosis	
	SOD1 G93A	-	
	BPA	- Drp1 translocation - mitochondrial RCS	
	BSSG	-	
Migraine	inflammatory soup	- small, fragmented mitochondria - reduced mt DNA - increased Drp1 fission protein - decreased Mfn1 fusion protein - valproic acid stabilized mitochondria	- decreased L-KYN, KYNA, 3-HK, 3-HAA, 5-HIAA, QUIN - increased L-Trp, AA, XA
	nitroglycerin-induced trigeminovascular activation	-	

**Table 3 cells-11-02607-t003:** The preclinical models, mitochondrial involvement, and findings in kynurenines in main psychiatric diseases.

Psychiatric Diseases	Preclinical Models	Mitochondrial Involvement	Findings in Kynurenines
Major depressive disorder	CMS	- decreased ATP, ATPase activity	- decreased Trp, KYN, KYNA - increased QUIN
	TST	- altered membrane potential	
	FST	-	
	Tph1^−/−^	-	
	Tph2^−/−^	-	
	Tph1/Tph2^−/−^	-	
	TPH2 variant (R439H) KI	-	
Generalized anxiety disorder	outbred Wistar rats	- reduced mitochondrial GTPase expression - altered mitochondrial morphology and functions	- decresed KYN
	social hierarchy	- NAc mitochondrial bioenergetic profiles	
Post-traumatic stress disorder	FKBP5^−/−^	-	-
	PAC1R^−/−^	-	
	5-HT1AR^−/−^	-	
	COMT^−/−^	-	
	GAD6^−/−^	-	
	GABAB1a^−/−^	-	
	CB1R^−/−^	-	
	single prolonged stress model	- abnormal apoptosis	
Bipolar disorder	ClockΔ19	-	- reduced KYNA - increased 3-HK/KYN, 3-HK/KYNA ratio - increased KYNA in CSF
	dominant negative mutant of mtDNA Polg1	-	
	-	- complex I expression abnormality	
Substance use disorder	-	- reduced mitochondrial copy numbers	- higher 5-HT - lower KYN/5-HT ratio
Schizophrenia	DISC1	- affect mitochondrial transport, fission, and fusion	- higher KYN and KYN/TRP ratio - increased KYN, KYNA - decreased KYNA/KYN ratio
	hypertensive rats	-	
Autism spectrum disorder	ND6^P25L^KI	-	- lowered KYNA - higher KYN/KYNA ratio - higher KYN/Trp ration, KYN, QUIN
	Shank3^Δc/Δc^	-	
	Cntnap2 KO	-	
	ADGRL3^−/−^	-	
	valproate	-	
	polyinosinic–polycytidylic acid	- mitochondrial dysfunction	
Attention-deficit hyperactivity disorder	Ptchd1^−/−^	-	- lowered Trp, KYNA, XA, 3-HAA - higher Trp, KYN
	-	- higher mtDNA copy number	

## Data Availability

Not applicable.

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
