# Peer review of "Mitochondrial Impairment: A Common Motif in Neuropsychiatric Presentation? The Link to the Tryptophan–Kynurenine Metabolic System"

_cells, 2022, doi:10.3390/cells11162607_

Round 1

Reviewer 1 Report

This is a very comprehensive review covering mitochondria and the TRP-KYN system in neurological and psyichiatric disorders. 

The majority of points are in spelling and grammar and use of some more common expressions in English. These can be found in the annotated PDF file that is attached. 

There are only 2 figures in the manuscript, however, the second is labelled figure 3. This figure does not display a link between the cytoplasmic and mitochondrial metabolic pathways. This should be included in the figure. 

Author Response

Reviewer 1:

This is a very comprehensive review covering mitochondria and the TRP-KYN system in neurological and psyichiatric disorders. 

The majority of points are in spelling and grammar and use of some more common expressions in English. These can be found in the annotated PDF file that is attached. 

There are only 2 figures in the manuscript, however, the second is labelled figure 3. This figure does not display a link between the cytoplasmic and mitochondrial metabolic pathways. This should be included in the figure. 

Response: Thank you for your valuable review report. A graphical abstract has been added. We all hope that it helps the reader grasp the essence of this manuscript. In addition, three tables have been added to summarize the sections. Regarding Figure 2, we intend to display and thus draw attention to the fact that the enzymes of the tryptophan-kynurenine (KYN) metabolic system are in a close contact with mitochondria, such as kynurenine 3-monooxygenase (KMO) which is located in the outer membrane of mitochondria and the isoforms of kynurenine aminotransferases (KATs), some of which are located in the matrix or the inner membrane of mitochondria. Presumably, KYN metabolites freely cross the mitochondria membranes. This is not depicted in the figure but added in the caption.  Finally, we sincerely appreciate your valuable time and kind help, providing us with the PDF annotating the comments and suggestions.

Minor errors have been corrected or rephrased as suggested. The following passages were added accordingly: 

“Mitophagy is induced by prolonged fission, promoting repair process but may lead to mitochondrial degradation.”

“IDO1 also catalyzes a stereoisomer D-Trp which is a product of the gut microbiome.”

Reviewer 2 Report

The article entitled “Mitochondrial Impairment: A Common Motif in Neuropsychiatric Presentation? The Link to the Tryptophan-Kynurenine Metabolic System” reviews the involvement of mitochondrial impairment in major neurological and psychiatric disorders. Comments and some of the language errors are listed below. After corrections, the manuscript should be suitable for publication in the Cells Journal. 

1.     The authors indicate in the abstract and the introduction that the article is a narrative review, which it is not. The narrative reviews follow a specific scheme including the description of the search methodology, inclusion criteria, etc. Please rethink the choice of article type or adjust the manuscript to the guidelines of narrative reviews or do not use the phrase “narrative review”. 

2.     line 57 and 76: did the authors mean aging of the cell/organism/mitochondria? please specify.

3.     line 59: “direct influence of inflammation… is misleading; did you mean maybe metabolites taking part in listed processes? please specify.

4.     lines 55-62: this paragraph might be elaborated on the basic functions of the Trp-Kyn system in the organism, as the authors have chosen to describe in the Introduction basis of mitochondrial function, it should also be added to the Trp-Kyn system description.

5.     lines 63-66: in the current form this fragment indicates that the mitochondrial dysfunction results only from the impairment in cellular energy production, which is not correct. Please correct this fragment and eliminate misleading information. 

6.      line 79: “due to oxidative stress” should be corrected as it is not quite exact. damage to mtDNA is caused by e.g., ROS which in mitochondria are produced not only during oxidative stress. 

7.     The Introduction should be reorganized. Try to focus first on the mitochondria and then on the Trp-Kyn system to enhance the flow of the introduction and engage the reader. Moreover, the Introduction repeats much of the information presented later in the text. This is not a goal of the Introduction, so please rethink this paragraph carefully. Another option is to delete chapter 2 and focus on the mitochondria and a short description of their basic functions in the introduction. 

8.     line 94: please specify the type of cell mentioned.

9.     lines 102-143: This fragment is unnecessary, please remove it. If authors want to keep such basic information about mitochondria, they should choose one form. The same is shown in the figure, there is no need to repeat information in the text and the figure. 

10.  lines 173-175: please remove the sentence “Lysozymes digest … cellular membrane”.

11.  Chapter 2: unfortunately, this section seems to be a cluster of random information without any flow or connection made by the authors thus, it is not interesting to read. Please rethink the narrative or remove it. (on the other hand, the next chapter is well-written, easy to follow, and interesting for the reader; it includes all necessary information, and the connection between the Trp-Kyn system and mitochondrial dysfunctions is indicated). Also, I suggest moving Figure 1 to chapter 3 where it is mentioned and relevant.

12.  Figure 3: please DO NOT copy the figure caption from the text. It should be relevant to the figure and not repeated with the text. Also, the figure numbering is not correct.

13.  Chapter 3: please remove/amend and keep to a minimum all information that is presented in the figure not to repeat it and elongate the manuscript more than necessary. 

14.  line 383: which “highly reactive free radical” do authors have in mind? please specify.

15.  line 402: research or researchers? please change to either “research is focused” or “researchers are focusing”

16.  Chapter 4: all diseases or neurological diseases? please specify the title of the chapter

17.  line 421: it would be worth mentioning why is that and what determines the localization of mitochondria in specific compartments of neural cells. 

18.  lines 426-438: please focus in this paragraph on the diseases connected to the nervous system, as it is the focus of the article. 

19.  lines 435-438 and line 471: If authors mention “there is no cure for mitochondria diseases” they should mention novel possibilities for therapies that are considered or in clinical trials. 

20.  line 468 and ref. 144: Such a statement is incorrect. There are more recent studies revealing the possibility of paternal inheritance of mtDNA. https://www.pnas.org/doi/10.1073/pnas.1810946115 
The authors should at least mention it or avoid such a strong statement about maternal inheritance patterns. 

21.  line 499: please remove the subtitle 4.3. as the whole article focuses on neurological diseases and it is clear at this point. such title should be placed as the main title of chapter 4

22.  line 514: if not to AD, then to what is it a characteristic?

23.  line 549: please specify the species of individuals

24.  line 825: please change “percent” to %

English language: 

line 42: please remove “which is directly usable to the host cells”

line 45 and 408: please remove “among others” (there already is “including” used in the sentences which indicate that only a few possibilities are mentioned. 

line 53: please change to “serving as signaling molecules in fundamental…”

line 66: 1 per/in 5000 births?

line 71: please change to “caused by the genes encoding for mtDNA transcription…”

line 96: “mitochondria are”

line 171: “through” is repeated

line 176: “ions”

line 194: MOMP etc. – do not introduce abbreviations that are used in the text only once.

line 455: “is characterized by neurological involvement of infancy or childhood” is misleading and not quite correctly put, please rewrite it to make it more clear for the reader. 

line 479: “Trp can metabolize”

line 494: please remove “and is in contrast to PMD, which can only be inherited”

line 508: “motor” is repeated twice

Author Response

Reviewer 2:

The article entitled “Mitochondrial Impairment: A Common Motif in Neuropsychiatric Presentation? The Link to the Tryptophan-Kynurenine Metabolic System” reviews the involvement of mitochondrial impairment in major neurological and psychiatric disorders. Comments and some of the language errors are listed below. After corrections, the manuscript should be suitable for publication in the Cells Journal. 

Response: Thank you for your valuable review report and kind endorsement for publication. We addressed our responses below and revised the manuscript accordingly.

  1. The authors indicate in the abstract and the introduction that the article is a narrative review, which it is not. The narrative reviews follow a specific scheme including the description of the search methodology, inclusion criteria, etc. Please rethink the choice of article type or adjust the manuscript to the guidelines of narrative reviews or do not use the phrase “narrative review”. 

Response: We removed “narrative” from the manuscript.

  1. line 57 and 76: did the authors mean aging of the cell/organism/mitochondria? please specify.

Response: We clarified ageing as follows: “normal ageing in organisms”.

  1. line 59: “direct influence of inflammation… is misleading; did you mean maybe metabolites taking part in listed processes? please specify.

Response: To clarify the message, the passage has been rephrased to: “The enzymes of the Trp-KYN system are activated by inflammation…”

  1. lines 55-62: this paragraph might be elaborated on the basic functions of the Trp-Kyn system in the organism, as the authors have chosen to describe in the Introduction basis of mitochondrial function, it should also be added to the Trp-Kyn system description.

Response: We tried to elaborate the paragraph by adding the following passage to the beginning: “Meanwhile, the tryptophan (Trp)-kynurenine (KYN) metabolic system plays a major role in Trp metabolism: over 95% of Trp catabolizes into nicotinamide adenine dinucleotide (NADH).” We hope that it suffices to make it clear.

  1. lines 63-66: in the current form this fragment indicates that the mitochondrial dysfunction results only from the impairment in cellular energy production, which is not correct. Please correct this fragment and eliminate misleading information. 

Response: We added the following to the end of passage: “… and other crucial mitochondrial functions.” We hope that this avoids the confusion.

  1. line 79: “due to oxidative stress” should be corrected as it is not quite exact. damage to mtDNA is caused by e.g., ROS which in mitochondria are produced not only during oxidative stress. 

Response: The end of the passage was revised by replacing it with “by reactive chemical species”.

  1. The Introduction should be reorganized. Try to focus first on the mitochondria and then on the Trp-Kyn system to enhance the flow of the introduction and engage the reader. Moreover, the Introduction repeats much of the information presented later in the text. This is not a goal of the Introduction, so please rethink this paragraph carefully. Another option is to delete chapter 2 and focus on the mitochondria and a short description of their basic functions in the introduction. 

Response: We tried to shorten the Introduction to avoid iteration of contents in the following sections. The end of the first paragraph was abridged as follows: “Other functions of mitochondria include calcium storage, subcellular signaling such as gene expression, autophagy, and apoptosis, among others [14-15].” The last passage of the second paragraph was removed.

  1. line 94: please specify the type of cell mentioned.

Response: The tissue type has been specified as follows: ”The estimated number of one to two million mitochondria is present per single neuron of the human substantia nigra [28]”.

  1. lines 102-143: This fragment is unnecessary, please remove it. If authors want to keep such basic information about mitochondria, they should choose one form. The same is shown in the figure, there is no need to repeat information in the text and the figure. 

Response: We are aware that the section resembles a biochemistry textbook. Our purpose of this section is to describe a tip of historical work of Krebs, Lippmann, and Szent-Györgyi to which the authors are affiliated and change in the cycle in the pathological state, beside capitulating the TCA cycle. We hope that the section serves as the physiological description of mitochondrial bioenergetics.   

  1. lines 173-175: please remove the sentence “Lysozymes digest … cellular membrane”.

Response: It was removed.

  1. Chapter 2: unfortunately, this section seems to be a cluster of random information without any flow or connection made by the authors thus, it is not interesting to read. Please rethink the narrative or remove it. (on the other hand, the next chapter is well-written, easy to follow, and interesting for the reader; it includes all necessary information, and the connection between the Trp-Kyn system and mitochondrial dysfunctions is indicated). Also, I suggest moving Figure 1 to chapter 3 where it is mentioned and relevant.

Response: We can believe that this section is not informative at all for mitochondria experts. Our intention with this section is to briefly summarize the functions of mitochondria including energy production and other functions for those who are not so familiar with mitochondria such as clinicians. We would be pleased to hear the sequence and the contents to be added in order to attract more readers.

  1. Figure 3: please DO NOT copy the figure caption from the text. It should be relevant to the figure and not repeated with the text. Also, the figure numbering is not correct.

Response: The caption is revised as follows and the numbering is corrected to Figure 2: “… and the metabolites freely cross the mitochondrial membranes.”.

  1. Chapter 3: please remove/amend and keep to a minimum all information that is presented in the figure not to repeat it and elongate the manuscript more than necessary. 

Response: All abbreviations are expanded as the names of the metabolites and the enzymes are presented with abbreviations in the figure.

  1. line 383: which “highly reactive free radical” do authors have in mind? please specify.

Response: It was corrected as follows: “a highly reactive hydroxyl free radical”.

  1. line 402: research or researchers? please change to either “research is focused” or “researchers are focusing”

Response: It is corrected as follows: “… researchers are focusing …”.

  1. Chapter 4: all diseases or neurological diseases? please specify the title of the chapter

Response: The tile of the Section 4 is about diseases in general. Then, it is followed by primary and secondary mitochondrial diseases in brief and neurological and psychiatric diseases in detail.

  1. line 421: it would be worth mentioning why is that and what determines the localization of mitochondria in specific compartments of neural cells. 

Response: The following passage was added: “…, suggesting most energy consumption takes place at the postsynaptic side”.

  1. lines 426-438: please focus in this paragraph on the diseases connected to the nervous system, as it is the focus of the article. 

Response: This manuscript focuses on neurological diseases and psychiatric disorders. Nevertheless, we believe that it is very important to take a broader view on mitochondrial-related diseases including comorbidity which often present neurological and/or psychiatric symptoms. So, we take a privilege to present it here before getting into details.

  1. lines 435-438 and line 471: If authors mention “there is no cure for mitochondria diseases” they should mention novel possibilities for therapies that are considered or in clinical trials. 

Response: The following passages were added with a reference: “Nevertheless, novel treatment for mitochondrial diseases is under extensive research. The strategies include oxidative stress modulation, mitochondrial biogenesis augmentation, mitochondrial autophagy modulation, nitric oxide restoration, mitochondria genome modulation, nucleotides pool restoration, hypoxia, enzyme replacement, and mitochondrial augmentation.”

  1. line 468 and ref. 144: Such a statement is incorrect. There are more recent studies revealing the possibility of paternal inheritance of mtDNA. https://www.pnas.org/doi/10.1073/pnas.1810946115 
    The authors should at least mention it or avoid such a strong statement about maternal inheritance patterns. 

Response: Thank you for your insightful comment with a reference. The passage was revised accordingly: “The mutations of mtDNA are considered to be inherited only maternally; however, a biparental mode of inheritance of mtDNA has been reported”.

  1. line 499: please remove the subtitle 4.3. as the whole article focuses on neurological diseases and it is clear at this point. such title should be placed as the main title of chapter 4

Response: The Chapter 4 consists of primary mitochondrial diseases, secondary mitochondrial dysfunctions, neurological diseases linked to mitochondrial impairments, and psychiatric disorders linked to mitochondrial dysfunction. The first two subsections are briefly documented as it is not a main purpose of this manuscript, but the subsections serve as solid clinical evidence in mitochondrial impairment. So, it is indispensable. The link of mitochondrial dysfunction to neuropsychiatric diseases remains not so solid clinically, but preclinical research has been presenting much more solid evidence. We intend to present it as a core part of this manuscript.

  1. line 514: if not to AD, then to what is it a characteristic?

Response: The passage was revised as follows: “The deposition of amyloid beta (Aβ) peptide and tau protein is a characteristic finding, but not limited to AD.”.

  1. line 549: please specify the species of individuals

Response: We have made it clear as follows: “Human studies in healthy individuals …”

  1. line 825: please change “percent” to %

Response: It was corrected.

English language: 

Response: Thank you for your careful reading. We revised them accordingly.

line 42: please remove “which is directly usable to the host cells”

line 45 and 408: please remove “among others” (there already is “including” used in the sentences which indicate that only a few possibilities are mentioned. 

line 53: please change to “serving as signaling molecules in fundamental…”

line 66: 1 per/in 5000 births?

line 71: please change to “caused by the genes encoding for mtDNA transcription…”

line 96: “mitochondria are”

line 171: “through” is repeated

line 176: “ions”

line 194: MOMP etc. – do not introduce abbreviations that are used in the text only once.

line 455: “is characterized by neurological involvement of infancy or childhood” is misleading and not quite correctly put, please rewrite it to make it more clear for the reader. 

line 479: “Trp can metabolize”

line 494: please remove “and is in contrast to PMD, which can only be inherited”

line 508: “motor” is repeated twice

Round 2

Reviewer 2 Report

A graphical abstract is the most welcome addition to the text. The authors have made minor corrections to the manuscript, which might be described as satisfactory, but not solid. Nevertheless, they did not make necessary changes to increase the focus of the article and make it less general and textbook-like. In my opinion, the strength of this review article is based on the description of the very particular connection between the Tryptophan-Kynurenine Metabolic System and Mitochondrial Impairment and it should stay as concise and focused as possible. Meanwhile, the authors have not changed the sections of the manuscript where some general information about mitochondria and other diseases is presented which are unnecessary if the focus of the manuscript is to be kept. 

Moreover, the text should be checked by a native speaker to correct the flow of the English language and shorten the unnecessarily long sentences without losing the meaning. Also, please correct the spelling of “ageing” which should be “aging”.

Author Response

A graphical abstract is the most welcome addition to the text. The authors have made minor corrections to the manuscript, which might be described as satisfactory, but not solid. Nevertheless, they did not make necessary changes to increase the focus of the article and make it less general and textbook-like. In my opinion, the strength of this review article is based on the description of the very particular connection between the Tryptophan-Kynurenine Metabolic System and Mitochondrial Impairment and it should stay as concise and focused as possible. Meanwhile, the authors have not changed the sections of the manuscript where some general information about mitochondria and other diseases is presented which are unnecessary if the focus of the manuscript is to be kept. 

We sincerely appreciate the reviewer’s continuous work for the betterment of our manuscript. We have tried to make our best effort to compile and present this manuscript covering mitochondria, the Krebs cycle, kynurenines, neurological diseases, and psychiatric disorders. We are all kynurenine researchers with different backgrounds: a clinical neurologist with long years of basic research experience, a biologist with molecular biology, a biologist in migraine research, a PhD student with biology background, and a medical doctor with preclinical depression research. We all understand the reviewer’s messages regarding very basic description of mitochondria and the Krebs cycle, which, indeed, might be unnecessary and even may off-focus this manuscript. Our main purpose of this manuscript is to make it readable to a wide range of scholars from basic scientists to family doctors, who may currently have various degrees of knowledge regarding contents in this manuscript. That said, I hope that one who is familiar with some contents just skip the part, while another may be able to refresh the basic knowledge and that all readers can eventually focus on the main message of this manuscript: the involvement of kynurenines in mitochondria for presentation of neuropsychiatric symptoms.

To make clear this purpose, we added the following passages in the ends of Section 2.1 and 2.2:

‘However, the cellular energy production can be altered under stressful condition or pathological processes according to availability of substrates, enzyme activity, mitochondrial and cellular conditions, and adjacent biosystems including Trp-KYN metabolic system.’

‘Therefore, mitochondria impairment may lead to multifarious consequences from ion homeostasis to entire organismal levels.’

Moreover, the text should be checked by a native speaker to correct the flow of the English language and shorten the unnecessarily long sentences without losing the meaning. Also, please correct the spelling of “ageing” which should be “aging”.

The manuscript has been corrected by a certified English language proof-reader. All ‘ageing’ is corrected to an American style ‘aging’.